# DEEP SYMBOLIC REGRESSION:
# RECOVERING MATHEMATICAL EXPRESSIONS FROM DATA VIA RISK-SEEKING POLICY GRADIENTS

**Brenden K. Petersen**[*]
Lawrence Livermore National Laboratory
Livermore, CA, USA
bp@llnl.gov

**Mikel Landajuela Larma**
Lawrence Livermore National Laboratory
Livermore, CA, USA
landajuelala1@llnl.gov

**T. Nathan Mundhenk**
Lawrence Livermore National Laboratory
Livermore, CA, USA
mundhenk1@llnl.gov

**Claudio P. Santiago**
Lawrence Livermore National Laboratory
Livermore, CA, USA
santiago10@llnl.gov

**Soo K. Kim**
Lawrence Livermore National Laboratory
Livermore, CA, USA
kim79@llnl.gov

**Joanne T. Kim**
Lawrence Livermore National Laboratory
Livermore, CA, USA
kim102@llnl.gov

## ABSTRACT

Discovering the underlying mathematical expressions describing a dataset is a core challenge for artificial intelligence. This is the problem of *symbolic regression*. Despite recent advances in training neural networks to solve complex tasks, deep learning approaches to symbolic regression are underexplored. We propose a framework that leverages deep learning for symbolic regression via a simple idea: use a large model to search the space of small models. Specifically, we use a recurrent neural network to emit a distribution over tractable mathematical expressions and employ a novel risk-seeking policy gradient to train the network to generate better-fitting expressions. Our algorithm outperforms several baseline methods (including Eureqa, the gold standard for symbolic regression) in its ability to exactly recover symbolic expressions on a series of benchmark problems, both with and without added noise. More broadly, our contributions include a framework that can be applied to optimize hierarchical, variable-length objects under a black-box performance metric, with the ability to incorporate constraints in situ, and a risk-seeking policy gradient formulation that optimizes for best-case performance instead of expected performance.

## 1 INTRODUCTION

Understanding the mathematical relationships among variables in a physical system is an integral component of the scientific process. Symbolic regression aims to identify these relationships by searching over the space of tractable (i.e. concise, closed-form) mathematical expressions to best fit a dataset. Specifically, given a dataset $(X, y)$, where each point $X_i \in \mathbb{R}^n$ and $y_i \in \mathbb{R}$, symbolic regression aims to identify a function $f : \mathbb{R}^n \to \mathbb{R}$ that best fits the dataset, where the functional form of $f$ is a short mathematical expression. The resulting expression can be readily interpreted and/or provide useful scientific insights simply by inspection. In contrast, conventional regression imposes a single model structure that is fixed during training, often chosen to be expressive (e.g. a neural network) at the expense of being easily interpretable.

---

[*]Corresponding author.

Symbolic regression exhibits several unique features that make it an excellent test problem for benchmarking automated machine learning (AutoML) and program synthesis methods: (1) there exist well-established, challenging benchmark problems with stringent success criteria (White et al., 2013); (2) there exist well-established baseline methods (most notably, the Eureqa algorithm (Schmidt & Lipson, 2009)); and (3) the reward function is computationally expedient, allowing sufficient experiment replicates to achieve statistical significance. Most other AutoML tasks, e.g. neural architecture search (NAS), do not exhibit these features; in fact, even simply *evaluating* the efficiency of the discrete search itself is a known challenge within NAS (Yu et al., 2019).

The space of mathematical expressions is discrete (in model structure) and continuous (in model parameters), growing exponentially with the length of the expression, rendering symbolic regression a challenging machine learning problem—thought to be NP-hard (Lu et al., 2016). Given this large, combinatorial search space, traditional approaches to symbolic regression typically utilize evolutionary algorithms, especially genetic programming (GP) (Koza, 1992; Schmidt & Lipson, 2009; Bäck et al., 2018). In GP-based symbolic regression, a population of mathematical expressions is "evolved" using evolutionary operations like selection, crossover, and mutation to improve a fitness function. While GP can be effective, it is also known to scale poorly to larger problems and to exhibit high sensitivity to hyperparameters.

Deep learning has permeated almost all areas of artificial intelligence, from computer vision (Krizhevsky et al., 2012) to optimal control (Mnih et al., 2015). However, deep learning may seem incongruous with or even antithetical toward symbolic regression, given that neural networks are typically highly complex, difficult to interpret, and rely on gradient information. We propose a framework that resolves this incongruity by tying deep learning and symbolic regression together with a simple idea: use a large model (i.e. neural network) to search the space of small models (i.e. symbolic expressions). This framework leverages the representational capacity of neural networks to generate interpretable expressions, while entirely bypassing the need to interpret the network itself.

We present *deep symbolic regression* (DSR), a gradient-based approach for symbolic regression based on reinforcement learning. In DSR, a recurrent neural network (RNN) emits a distribution over mathematical expressions. Expressions are sampled from the distribution, instantiated, and evaluated based on their fitness to the dataset. This fitness is used as the reward signal to train the RNN using a novel risk-seeking policy gradient algorithm. As training proceeds, the RNN adjusts the likelihood of an expression relative to its reward, assigning higher probabilities to better expressions.

We demonstrate that DSR outperforms several baseline methods, including two commercial software algorithms. We summarize our contributions as follows: (1) a novel method for symbolic regression that outperforms several baselines on a set of benchmark problems, (2) an autoregressive generative modeling framework for optimizing hierarchical, variable-length objects that accommodates in situ constraints, and (3) a novel risk-seeking policy gradient objective and accompanying Monte Carlo estimation procedure that optimizes for best-case performance instead of average performance.

## 2 RELATED WORK

**Deep learning for symbolic regression.** Several recent approaches leverage deep learning for symbolic regression. AI Feynman (Udrescu & Tegmark, 2020) propose a problem-simplification tool for symbolic regression. They use neural networks to identify simplifying properties in a dataset (e.g. multiplicative separability, translational symmetry), which they exploit to recursively define simplified sub-problems that can then be tackled using any symbolic regression algorithm. In GrammarVAE, Kusner et al. (2017) develop a generative model for discrete objects that adhere to a pre-specified grammar, then optimize them in latent space. They demonstrate this can be used for symbolic regression; however, the method struggles to exactly recover expressions, and the generated expressions are not always syntactically valid. Sahoo et al. (2018) develop a symbolic regression framework using neural networks whose activation functions are symbolic operators. While this approach enables an end-to-end differentiable system, backpropagation through activation functions like division or logarithm requires the authors to make several simplifications to the search space, ultimately precluding learning certain simple classes of expressions like $\sqrt{x}$ or $\sin(x/y)$. We address and/or directly compare to these works in Appendices C and E.

**AutoML and program synthesis.** Symbolic regression is related to both automated machine learning (AutoML) and program synthesis, in that they all involve a search for an executable program (i.e. expression) to solve a particular task (i.e. to fit data) (Abolafia et al., 2018; Devlin et al., 2017; Riedel et al., 2016). More specifically, our framework has many parallels to a body of works within AutoML that use an autoregressive RNN to define a distribution over discrete objects and use reinforcement learning to optimize this distribution under a black-box performance metric (Zoph & Le, 2017; Ramachandran et al., 2017; Bello et al., 2017; Abolafia et al., 2018). For example, in neural architecture search (Zoph & Le, 2017), an RNN searches the space of neural network architectures, encoded by a sequence of discrete "tokens" specifying architectural properties (e.g. number of neurons) of each layer. The length of the sequence is fixed or scheduled during training; in contrast, our framework defines a search space that is both inherently hierarchical and variable length. Ramachandran et al. (2017) search the space of neural network activation functions. While this space is hierarchical in nature, the authors (rightfully) constrain it substantially by positing a functional unit that is repeated sequentially, thus restricting their search space back to a fixed-length sequence. However, a repeating-unit constraint is not practical for symbolic regression because the ground truth expression may have arbitrary structure.

**Autoregressive models.** The RNN-based distribution over expressions used in DSR is autoregressive, meaning each token is conditioned on the previously sampled tokens. Autoregressive models have proven to be useful for audio and image data (Oord et al., 2016a;b) in addition to the AutoML works discussed above; we further demonstrate their efficacy for hierarchical expressions. GraphRNN defines a distribution over graphs that generates an adjacency matrix one column at a time in autoregressive fashion (You et al., 2018). In principle, GraphRNN could be constrained to define a distribution over expressions, since trees are a special case of graphs. However, GraphRNN constructs graphs breadth-first, whereas expressions are more naturally represented using depth-first traversals (Li et al., 2005). Further, DSR exploits the hierarchical nature of trees by providing the parent and sibling as inputs to the RNN, and leverages the additional structure of expression trees that a node's value determines its number of children (e.g. cosine is a unary operator and thus has one child).

**Risk-aware reinforcement learning.** Many of the AutoML methods discussed above suffer from what we call the "expectation problem." That is, policy gradient methods are fundamentally suited for optimizing *expectations*; however, domains like neural architecture search and symbolic regression are evaluated by the few or single best-performing samples. Thus, there is a mismatch between the training objective function and the true desired objective: to maximize best-case performance. Abolafia et al. (2018) address the expectation problem by maintaining a priority queue of the best seen samples and using supervised learning to increase the likelihood of those top samples. Similarly, Liang et al. (2018) use a memory buffer to augment a policy gradient with off-policy training. These methods, however, only apply in the context of reinforcement learning environments with both deterministic transition dynamics and deterministic rewards. In contrast, the *risk-seeking policy gradient* introduced here is general, applying to any reinforcement learning environment and any stochastic policy gradient algorithm trained using batches, e.g. on Atari using proximal policy optimization (Schulman et al., 2017). Lastly, our risk-seeking policy gradient is closely related to the EPOpt-$\varepsilon$ algorithm used for robust reinforcement learning (Rajeswaran et al., 2016), which is based on a risk-averse policy gradient formulation (Tamar et al., 2014).

## 3 METHODS

Our overall approach involves representing mathematical expressions as sequences, developing an autoregressive model to generate expressions under a pre-specified set of constraints, and developing a risk-seeking policy gradient to train the model to generate better-fitting expressions.

### 3.1 GENERATING EXPRESSIONS WITH A RECURRENT NEURAL NETWORK

We leverage the fact that mathematical expressions can be represented using *symbolic expression trees*. Expression trees are a type of binary tree in which internal nodes are mathematical operators and terminal nodes are input variables or constants. Operators may be unary (i.e. one argument, such as sine) or binary (i.e. two arguments, such as multiply). Further, we can represent an expression tree as a sequence of node values or "tokens" by using its pre-order traversal (i.e. by visiting each node depth-first, then left-to-right). This allows us to generate an expression tree sequentially while still

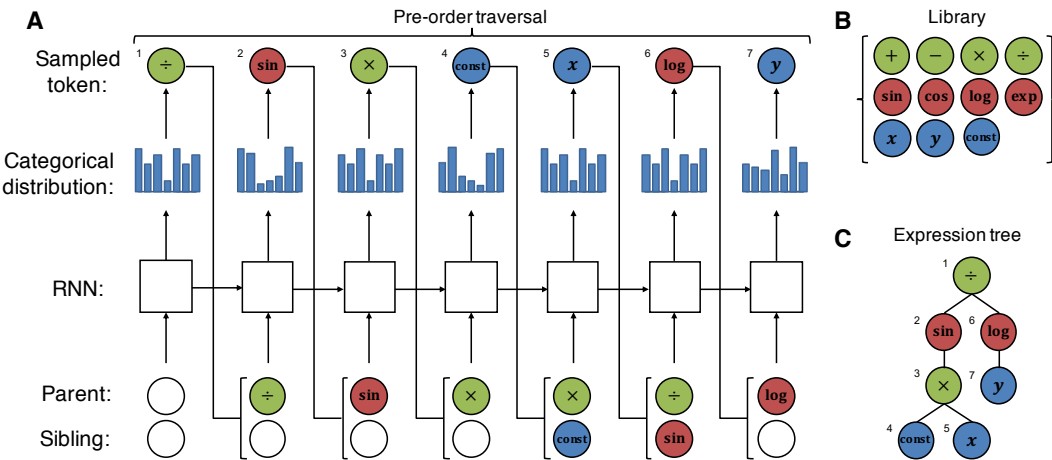

Figure 1: **A**. Example of sampling an expression from the RNN. For each token, the RNN emits a categorical distribution over tokens, a token is sampled, and the parent and sibling of the next token are used as the next input to the RNN. Subsequent tokens are sampled autoregressively until the tree is complete (i.e. all tree branches reach terminal nodes). The resulting sequence of tokens is the tree's pre-order traversal, which can be used to reconstruct the tree and instantiate its corresponding expression. Colors correspond to the number of children for each token. White circles represent empty tokens. Numbers indicate the order in which tokens were sampled. **B**. The library of tokens. **C**. The expression tree sampled in **A**. In this example, the sampled expression is $\sin(cx)/\log(y)$, where the value of the constant $c$ is optimized with respect to an input dataset.

maintaining a one-to-one correspondence between the tree and its traversal.[1] Thus, we represent an expression $\tau$ by the pre-order traversal of its corresponding expression tree.[2] We denote the $i^{\text{th}}$ token of the traversal as $\tau_i$ and the length of the traversal as $|\tau| = T$. Each token has a value within a given library $\mathcal{L}$ of possible tokens, e.g. $\{+, -, \times, \div, \sin, x\}$.

We generate expressions one token at a time along the pre-order traversal (from $\tau_1$ to $\tau_T$). A categorical distribution with parameters $\psi$ defines the probabilities of selecting each token from $\mathcal{L}$. To capture the "context" of the expression as it is being generated, we condition this probability upon the selections of all previous tokens in that traversal. This conditional dependence can be achieved very generally using an RNN with parameters $\theta$ that emits a probability vector $\psi$ in an autoregressive manner. Specifically, the $i^{\text{th}}$ output of the RNN passes through a softmax layer (with shared weights across time steps) to produce vector $\psi^{(i)}$, which defines the probability distribution for selecting the $i^{\text{th}}$ token $\tau_i$, conditioned on the previously selected tokens $\tau_{1:(i-1)}$. That is, $p(\tau_i|\tau_{1:(i-1)}; \theta) = \psi_{\mathcal{L}(\tau_i)}^{(i)}$, where $\mathcal{L}(\tau_i)$ is the index in $\mathcal{L}$ corresponding to $\tau_i$. The likelihood of the entire sampled expression is simply the product of the likelihoods of its tokens: $p(\tau|\theta) = \prod_{i=1}^{|\tau|} p(\tau_i|\tau_{1:(i-1)}; \theta) = \prod_{i=1}^{|\tau|} \psi_{\mathcal{L}(\tau_i)}^{(i)}$.

An example of the sampling process is illustrated in Figure 1; pseudocode is provided in Algorithm 2 in Appendix A. Note that different samples from the distribution have different tree structures of different size; thus, the search space is inherently both hierarchical and variable length.

**Providing hierarchical inputs to the RNN.** Conventionally, the input to the RNN when sampling a token would be a representation of the previously sampled token. However, the search space for symbolic regression is inherently hierarchical, and the previously sampled token may actually be very distant from the next token to be sampled in the expression tree. For example, the fifth and

---

[1]In general, a pre-order traversal is insufficient to uniquely reconstruct the tree. However, here we know how many children each node has based on its value, e.g. "multiply" is a binary operator and thus has two children. A pre-order traversal plus the corresponding number of children for each node is sufficient to uniquely reconstruct the tree.

[2]Given an expression tree, the corresponding mathematical expression is unique; however, given an expression, its expression tree is not unique. For example, $x^2$ and $x \times x$ are equivalent expressions but yield different trees. For simplicity, we use $\tau$ somewhat abusively to refer to an expression where it technically refers to an expression tree (or equivalently, its pre-order traversal).

sixth tokens sampled in Figure 1 are adjacent nodes in the traversal but are four edges apart in the expression tree. To better capture hierarchical information, we provide as inputs to the RNN a representation of the parent and sibling nodes of the token being sampled. We introduce an empty token for cases in which a node does not have a parent or sibling. Pseudocode for identifying the parent and sibling nodes given a partial traversal is provided in Subroutine 1 in Appendix A.

**Constraining the search space.** Under our framework, it is straightforward to apply a priori constraints to reduce the search space. To demonstrate, we impose several simple, domain-agnostic constraints: (1) Expressions are limited to a pre-specified minimum and maximum length. We selected minimum length of 4 to prevent trivial expressions and a maximum length of 30 to ensure expressions are tractable. (2) The children of an operator should not all be constants, as the result would simply be a different constant. (3) The child of a unary operator should not be the inverse of that operator, e.g. $\log(\exp(x))$ is not allowed. (4) Descendants of trigonometric operators should not be trigonometric operators, e.g. $\sin(x + \cos(x))$ is not allowed because cosine is a descendant of sine. While still semantically meaningful, such composed trigonometric operators do not appear in virtually any scientific domain.

We apply these constraints in situ (i.e. concurrently with autoregressive sampling) by zeroing out the probabilities of selecting tokens that would violate a constraint. Pseudocode for this process is provided in Subroutine 2 in Appendix A. This process ensures that samples always adhere to all constraints, without rejecting samples post hoc. In contrast, imposing constraints with GP requires rejecting evolutionary operations post hoc (Fortin et al., 2012), which can be problematic (Craenen et al., 2001), and as we show in our experiments, can actually reduce performance.

**Reward function.** Once a pre-order traversal is sampled, we instantiate the corresponding symbolic expression and evaluate it with a reward function. A standard fitness measure in GP-based symbolic regression is normalized root-mean-square error (NRMSE), the root-mean-square error normalized by the standard deviation of the target values, $\sigma_y$. That is, given a dataset $(X, y)$ of size $n$ and candidate expression $f$, $\text{NRMSE} = \frac{1}{\sigma_y}\sqrt{\frac{1}{n}\sum_{i=1}^{n}(y_i - f(X_i))^2}$. To bound the reward function, we apply a squashing function: $R(\tau) = 1/(1 + \text{NRMSE})$.

**Constant optimization.** If the library $\mathcal{L}$ includes the constant token, sampled expressions may include several constant placeholders. These can be viewed as parameters $\xi$ of the symbolic expression, which we optimize by maximizing the reward function: $\xi^{\star} = \arg\max_{\xi} R(\tau; \xi)$, using a nonlinear optimization algorithm, e.g. BFGS (Fletcher, 2013). We perform this inner optimization loop for each sampled expression as part of the reward computation before performing each training step.

### 3.2 Training the RNN using policy gradients

The reward function is not differentiable with respect to $\theta$; thus, we turn to reinforcement learning to train the RNN to produce better-fitting expressions. In this view, the distribution over mathematical expressions $p(\tau|\theta)$ is like a policy, sampled tokens are like actions, the parent and sibling inputs are like observations, sequences corresponding to expressions are like episodes, and the reward is a terminal, undiscounted reward only computed when an expression completes.

**Standard policy gradient.** We first consider the standard policy gradient objective to maximize $J_{\text{std}}(\theta)$, defined as the expectation of a reward function $R(\tau)$ under expressions from the policy: $J_{\text{std}}(\theta) \doteq \mathbb{E}_{\tau \sim p(\tau|\theta)}[R(\tau)]$. The standard REINFORCE policy gradient (Williams, 1992) can be used to maximize this expectation via gradient ascent:

$$\nabla_\theta J_{\text{std}}(\theta) = \nabla_\theta \mathbb{E}_{\tau \sim p(\tau|\theta)}[R(\tau)]$$
$$= \mathbb{E}_{\tau \sim p(\tau|\theta)}[R(\tau)\nabla_\theta \log p(\tau|\theta)]$$

This result allows one to estimate the expectation using samples from the distribution. Specifically, an unbiased estimate of $\nabla_\theta J_{\text{std}}(\theta)$ can be obtained by computing the sample mean over a batch of $N$ sampled expressions $\mathcal{T} = \{\tau^{(i)}\}_{i=1}^{N}$:

$$\nabla_\theta J_{\text{std}}(\theta) \approx \frac{1}{N}\sum_{i=1}^{N} R(\tau^{(i)})\nabla_\theta \log p(\tau^{(i)}|\theta)$$

This is an unbiased gradient estimate, but in practice it has high variance. To reduce variance, it is common to subtract a baseline function $b$ from the reward. As long as the baseline is not a function of

the current batch of expressions, the gradient estimate is still unbiased. Common choices of baseline functions are a moving average of rewards or an estimate of the value function. Intuitively, the gradient step increases the likelihood of expressions above the baseline and decreases the likelihood of expressions below.

**Risk-seeking policy gradient.** The standard policy gradient objective, $J_{\text{std}}(\theta)$, is defined as an *expectation*. This is the desired objective for control problems in which one seeks to optimize the average performance of a policy. However, in domains like symbolic regression, program synthesis, or neural architecture search, the final performance is measured by the single or few best-performing samples found during training. Similarly, one might be interested in a policy that achieves a "high score" in a control environment (e.g. Atari). For such problems, $J_{\text{std}}(\theta)$ is not an appropriate objective, as there is a mismatch between the objective being optimized and the final performance metric; this is the "expectation problem." To address this disconnect, we propose an alternative objective that focuses learning only on *maximizing best-case performance*. We first define $R_\varepsilon(\theta)$ as the $(1-\varepsilon)$-quantile of the distribution of rewards under the current policy. Note this is a function of $\theta$ but is typically intractable. We then propose a new learning objective, $J_{\text{risk}}(\theta; \varepsilon)$, parameterized by $\varepsilon$:

$$J_{\text{risk}}(\theta; \varepsilon) \doteq \mathbb{E}_{\tau \sim p(\tau|\theta)} \left[ R(\tau) \mid R(\tau) \geq R_\varepsilon(\theta) \right] \tag{1}$$

This objective aims to increase the reward of the top $\varepsilon$ fraction of samples from the distribution, without regard for samples below that threshold. This objective bears close resemblance with $\varepsilon$-conditional value at risk (CVaR), for which the "$\leq$" symbol is used instead of "$\geq$" and the $\varepsilon$-quantile of rewards is used instead of the $(1-\varepsilon)$-quantile. Optimizing CVaR is a form of risk-averse learning that results in a policy that is robust against catastrophic outcomes (Tamar et al., 2014; Rajeswaran et al., 2016). In contrast, optimizing $J_{\text{risk}}(\theta; \varepsilon)$ yields a *risk-seeking* policy gradient that aims to increase best-case performance at the expense of lower worst-case and average performances. Next, we show the analogous policy gradient of $J_{\text{risk}}(\theta; \varepsilon)$ and how to estimate it via Monte Carlo sampling.

**Proposition 1.** *Let $J_{risk}(\theta; \varepsilon)$ denote the conditional expectation of rewards above the $(1-\varepsilon)$-quantile, as in Equation (1). Then the gradient of $J_{risk}(\theta; \varepsilon)$ is given by:*

$$\nabla_\theta J_{risk}(\theta; \varepsilon) = \mathbb{E}_{\tau \sim p(\tau|\theta)}[(R(\tau) - R_\varepsilon(\theta)) \cdot \nabla_\theta \log p(\tau|\theta) \mid R(\tau) \geq R_\varepsilon(\theta)]$$

The proof and assumptions are provided in Appendix B, and are adapted from the policy gradient derivation for the CVaR objective (Tamar et al., 2014). This result suggests a simple Monte Carlo estimate of the gradient from a batch of $N$ samples:

$$\nabla_\theta J_{\text{risk}}(\theta; \varepsilon) \approx \frac{1}{\varepsilon N} \sum_{i=1}^{N} \left[ R(\tau^{(i)}) - \tilde{R}_\varepsilon(\theta) \right] \cdot \mathbf{1}_{R(\tau^{(i)}) \geq \tilde{R}_\varepsilon(\theta)} \nabla_\theta \log p(\tau^{(i)}|\theta),$$

where $\tilde{R}_\varepsilon(\theta)$ is the empirical $(1-\varepsilon)$-quantile of the batch of rewards, and $\mathbf{1}_x$ returns 1 if condition $x$ is true and 0 otherwise. Essentially, this is the standard REINFORCE Monte Carlo estimate with two differences: (1) theory suggests a specific baseline, $R_\varepsilon(\theta)$, whereas the baseline for standard policy gradients is non-specific, chosen by the user; and (2) effectively, only the top $\varepsilon$ fraction of samples from each batch are used in the gradient computation. This process is essentially the opposite of the approach used to optimize CVaR (Tamar et al., 2014; Rajeswaran et al., 2016) for risk-averse reinforcement learning, in which only the *bottom* $\varepsilon$ fraction of samples from each batch are used.

Note that in Proposition 1 and the corresponding Monte Carlo estimation procedure, $\tau$ need not refer to a symbolic expression or even a discrete object. For example, it may refer to a state-action trajectory in a control problem. Thus, the risk-seeking policy gradient formulation is general and can easily be applied to any reinforcement learning environment using any stochastic policy gradient algorithm trained using batches, e.g. proximal policy optimization (Schulman et al., 2017).

Lastly, in accordance with the maximum entropy reinforcement learning framework (Haarnoja et al., 2018), we provide a bonus to the loss function proportional to the entropy of the sampled expressions. Pseudocode for DSR is shown in Algorithm 1. Source code is made available at `https://github.com/brendenpetersen/deep-symbolic-regression`.

## 4 RESULTS AND DISCUSSION

**Evaluating DSR.** We evaluated DSR on the Nguyen symbolic regression benchmark suite (Uy et al., 2011), a set of 12 commonly used benchmark expressions developed and vetted by the symbolic

---

**Algorithm 1** Deep symbolic regression with risk-seeking policy gradient

---

**input** learning rate $\alpha$; entropy coefficient $\lambda_{\mathcal{H}}$; risk factor $\varepsilon$; batch size $N$; reward function $R$
**output** Best fitting expression $\tau^{\star}$

1: Initialize RNN with parameters $\theta$, defining distribution over expressions $p(\cdot|\theta)$
2: **repeat**
3:     $\mathcal{T} \leftarrow \{\tau^{(i)} \sim p(\cdot|\theta)\}_{i=1}^{N}$              $\triangleright$ Sample $N$ expressions (Alg. 2 in Appendix A)
4:     $\mathcal{T} \leftarrow \{\text{OptimizeConstants}(\tau^{(i)}, R)\}_{i=1}^{N}$      $\triangleright$ Optimize constants w.r.t. reward function
5:     $\mathcal{R} \leftarrow \{R(\tau^{(i)})\}_{i=1}^{N}$                  $\triangleright$ Compute rewards
6:     $R_{\varepsilon} \leftarrow (1-\varepsilon)$-quantile of $\mathcal{R}$          $\triangleright$ Compute reward threshold
7:     $\mathcal{T} \leftarrow \{\tau^{(i)} : R(\tau^{(i)}) \geq R_{\varepsilon}\}$     $\triangleright$ Select subset of expressions above threshold
8:     $\mathcal{R} \leftarrow \{R(\tau^{(i)}) : R(\tau^{(i)}) \geq R_{\varepsilon}\}$     $\triangleright$ Select corresponding subset of rewards
9:     $\hat{g}_1 \leftarrow \text{ReduceMean}((\mathcal{R} - R_{\varepsilon})\nabla_{\theta} \log p(\mathcal{T}|\theta))$     $\triangleright$ Compute risk-seeking policy gradient
10:    $\hat{g}_2 \leftarrow \text{ReduceMean}(\lambda_{\mathcal{H}}\nabla_{\theta}\mathcal{H}(\mathcal{T}|\theta))$        $\triangleright$ Compute entropy gradient
11:    $\theta \leftarrow \theta + \alpha(\hat{g}_1 + \hat{g}_2)$                 $\triangleright$ Apply gradients
12:    **if** $\max \mathcal{R} > R(\tau^{\star})$ **then** $\tau^{\star} \leftarrow \tau^{(\arg\max \mathcal{R})}$     $\triangleright$ Update best expression
13: **return** $\tau^{\star}$

---

regression community (White et al., 2013). Each benchmark is defined by a ground truth expression, a training and test dataset, and a set of allowed operators, described in Table 2 in Appendix D. The training data is used to compute the reward for each candidate expression, the test data is used to evaluate the best found candidate expression at the end of training, and the ground truth expression is used to determine whether the best found candidate expression was correctly recovered.

We compared DSR against five strong symbolic regression baselines: (1) **PQT**: an implementation of our framework trained using priority queue training (Abolafia et al., 2018) in place of the risk-seeking policy gradient, (2) **VPG**: a "vanilla" implementation of our framework using the standard policy gradient (with baseline) in place of the risk-seeking policy gradient, (3) **GP**: a standard GP-based symbolic regression implementation (Fortin et al., 2012), (4) **Eureqa**: popular commercial software[3] based on Schmidt & Lipson (2009), the gold standard for symbolic regression, and (5) **Wolfram**: commercial software[4] based on Markov chain Monte Carlo and nonlinear regression (Fortuna, 2015). Each baseline is further detailed in Appendix C. Notably, the two RNN-based baselines (PQT and VPG) differ from DSR only via their training objective, i.e. a supervised objective for PQT and the standard policy gradient for VPG. All other aspects of our framework (in situ constraints, hierarchical RNN inputs, constant optimization) are also included for these baselines. While GP can incorporate post hoc constraints, we found that doing so actually hinders performance (see Appendix F); thus, they are excluded for GP we exclude all constraints other than the maximum length constraint.

Each experiment consists of 2 million expression evaluations for DSR, PQT, VPG, and GP, by which point training curves have levelled off. Hyperparameters were tuned on benchmarks Nguyen-7 and Nguyen-10 using a grid search comprising 800 hyperparameter combinations for GP and 81 combinations for each of DSR, PQT, and VPG (see Appendix D for details). Commercial software algorithms (Eureqa and Wolfram) do not expose hyperparameters and were run until completion. All experiments were replicated with 100 different random seeds for each benchmark expression. Wolfram results are only presented for one-dimensional benchmarks because the method is not applicable to higher dimensions. Additional experiment details are provided in Appendix D. Training curves are provided in Appendix F.

In Table 1, we report the recovery rate for each benchmark. We use the strictest definition of recovery: exact symbolic equivalence, as determined using a computer algebra system, e.g. SymPy (Meurer et al., 2017). In Table 9 in Appendix F, we report recovery on several additional variants of Nguyen benchmarks in which we introduced real-valued constants (to demonstrate the constant optimizer) or altered the functional form to make the problems more challenging. DSR significantly outperforms all five baselines in its ability to exactly recover benchmark expressions. In Tables 5–8 in Appendix E, we compare DSR to literature-reported values from four additional baseline methods (Huynh et al., 2016; Kusner et al., 2017; Jin et al., 2019; Trujillo et al., 2016) by carefully recapitulating their

---

[3]Sold by DataRobot, Inc. (www.datarobot.com) and formerly by Nutonian, Inc. (www.nutonian.com).
[4]Sold by Wolfram Research, Inc. as a part of Mathematica (www.wolfram.com/mathematica).

Table 1: Recovery rate comparison of DSR and five baselines on the Nguyen symbolic regression benchmark suite. A bold value represents statistical significance ($p < 10^{-3}$) across all benchmarks.

| Benchmark | Expression | DSR | PQT | VPG | GP | Eureqa | Wolfram |
|---|---|---|---|---|---|---|---|
| Nguyen-1 | $x^3 + x^2 + x$ | 100% | 100% | 96% | 100% | 100% | 100% |
| Nguyen-2 | $x^4 + x^3 + x^2 + x$ | 100% | 99% | 47% | 97% | 100% | 100% |
| Nguyen-3 | $x^5 + x^4 + x^3 + x^2 + x$ | 100% | 86% | 4% | 100% | 95% | 100% |
| Nguyen-4 | $x^6 + x^5 + x^4 + x^3 + x^2 + x$ | 100% | 93% | 1% | 100% | 70% | 100% |
| Nguyen-5 | $\sin(x^2)\cos(x) - 1$ | 72% | 73% | 5% | 45% | 73% | 2% |
| Nguyen-6 | $\sin(x) + \sin(x + x^2)$ | 100% | 98% | 100% | 91% | 100% | 1% |
| Nguyen-7 | $\log(x + 1) + \log(x^2 + 1)$ | 35% | 41% | 3% | 0% | 85% | 0% |
| Nguyen-8 | $\sqrt{x}$ | 96% | 21% | 5% | 5% | 0% | 71% |
| Nguyen-9 | $\sin(x) + \sin(y^2)$ | 100% | 100% | 100% | 100% | 100% | – |
| Nguyen-10 | $2\sin(x)\cos(y)$ | 100% | 91% | 99% | 76% | 64% | – |
| Nguyen-11 | $x^y$ | 100% | 100% | 100% | 7% | 100% | – |
| Nguyen-12 | $x^4 - x^3 + \frac{1}{2}y^2 - y$ | 0% | 0% | 0% | 0% | 0% | – |
| | Average | **83.6%** | 75.2% | 46.7% | 60.1% | 73.9% | – |

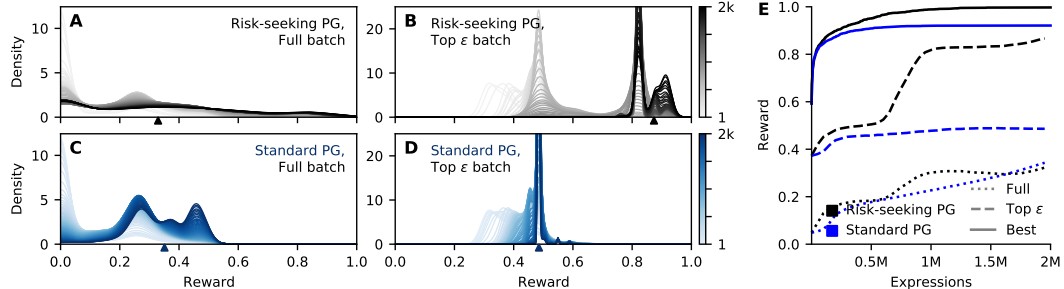

Figure 2: **A** - **D**. Empirical reward distributions for Nguyen-8. Each curve is a Gaussian kernel density estimate (bandwidth 0.25) of the rewards for a particular training iteration, using either the full batch of expressions (**A** and **C**) or the top $\varepsilon$ fraction of the batch (**B** and **D**), averaged over all training runs. Black plots (**A** and **B**) were trained using the risk-seeking policy gradient objective. Blue plots (**C** and **D**) were trained using the standard policy gradient objective. Colorbars indicate training step. Triangle markings denote the empirical mean of the distribution at the final training step. **E**. Training curves for mean reward of full batch (dotted), mean reward of top $\varepsilon$ fraction of the batch (dashed), and best expression found so far (solid), averaged over all training runs.

experimental setup (e.g. choice of library, benchmarks, and measure of performance). DSR greatly outperforms each study's published results.

**Characterizing the risk-seeking policy gradient.** The intuition behind the risk-seeking policy gradient is that it explicitly optimizes for best-case performance, possibly at the expense of average performance. We demonstrate this visually in Figure 2 by comparing the empirical reward distributions when trained with either the risk-seeking or standard policy gradient for Nguyen-8. (Analogous plots for all Nguyen benchmarks are provided in Appendix F.) Interestingly, at the end of training, the mean reward over the full batch (an estimate of $J_{\text{std}}(\theta)$) is larger when training with the standard policy gradient, even though the risk-seeking policy gradient produces larger mean over the top $\varepsilon$ fraction of the batch (an estimate of $J_{\text{risk}}(\theta; \varepsilon)$) and a superior best expression. This is consistent with the intuition of maximizing best-case performance at the expense of average performance. In contrast, the best-case performance of the standard policy gradient plateaus early in training (Figure 2E, dashed blue curve), whereas the risk-seeking policy gradient continues to increase until the end of training (Figure 2E, dashed black curve).

**Ablation studies.** Algorithm 1 includes several additional components relative to a "vanilla" policy gradient search. We performed a series of ablation studies to quantify the effect of each of these components, along with the effects of the various constraints on the search space. In Figure 3, we show recovery rate for DSR on the Nguyen benchmarks for each ablation. While no single ablation leads to catastrophic failure, combinations of ablations can cause large degradation in performance.

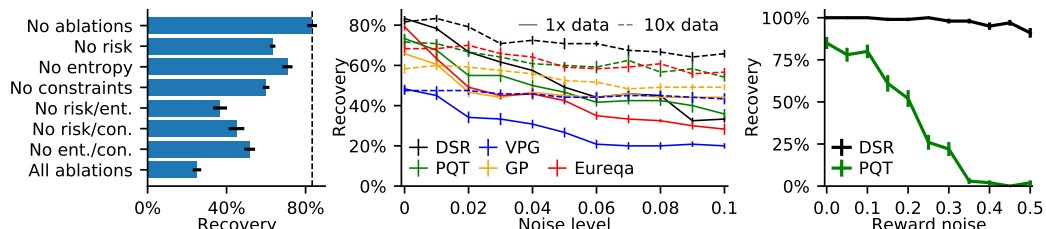

Figure 3: Recovery for various ablations of Algorithm 1 across all Nguyen benchmarks. Error bars represent standard error.

Figure 4: Recovery vs dataset noise and dataset size across all Nguyen benchmarks. Error bars represent standard error.

Figure 5: Recovery vs added reward noise on Nguyen-4. Error bars represent standard error.

**Noisy data and amount of data.** We evaluated the robustness of DSR to noisy data by adding independent Gaussian noise to the dependent variable, with mean zero and standard deviation proportional to the root-mean-square of the dependent variable in the training data. In Figure 4, we varied the proportionality constant from $0$ (noiseless) to $10^{-1}$ and evaluated each algorithm (except Wolfram, which catastrophically fails for even the smallest noise level) across all Nguyen benchmarks. Because expressions can overfit to the noise in these experiments, we defined recovery as exact symbolic equivalence on any expression along the reward-complexity Pareto front at the end of training (see Appendix D for details). Increasing the dataset size may help prevent overfitting by smoothing the reward function. Thus, we repeated the noise experiments with 10-fold larger training datasets (Figure 4, dashed lines). For each dataset size, DSR consistently outperforms all baselines.

**Performance under reward noise.** The risk-seeking policy gradient is derived from a conditional expectation, and is thus well-justified for tasks with stochastic rewards. In contrast, PQT assumes deterministic rewards. Since symbolic regression is a deterministic task, we emulated a stochastic reward function by adding independent Gaussian noise directly to the reward function: $R'(\tau) = R(\tau) + \mathcal{N}(0, \sigma)$. In Figure 5, we compare DSR and PQT performance under increasing reward noise on benchmark Nguyen-4, a task that is an easier exploitation problem but difficult exploration problem. As reward noise increases, recovery for PQT heavily relies on exploration and luck, whereas DSR recovery remains stable even for high reward noise.

## 5 CONCLUSION AND FUTURE WORK

We introduce a reinforcement learning approach to symbolic regression that outperforms state-of-the-art baselines in its ability to recover exact expressions on benchmark tasks. Since both DSR and GP generate expression trees, there are many opportunities for hybrid methods, for example including several generations of evolutionary operations within the inner optimization loop. Our framework includes a flexible distribution over hierarchical, variable-length objects that allows imposing in situ constraints. Our framework is easily extensible to other domains, which we save for future work; for example, searching the space of expressions to be used as control policies in reinforcement learning environments, or searching the space of organic molecular structures for high binding affinity to a reference compound. Our risk-seeking policy gradient formulation can also be applied to more traditional reinforcement learning domains; for example, optimizing for a high score (instead of average score) in Atari video games.

ACKNOWLEDGMENTS

We thank Ruben Glatt, Thomas Desautels, Priyadip Ray, David Widemann, and the 2019 UC Merced Data Science Challenge participants for their useful comments and insights. This work was performed under the auspices of the U.S. Department of Energy by Lawrence Livermore National Laboratory under contract DE-AC52-07NA27344. Lawrence Livermore National Security, LLC. LLNL-CONF-790457.

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

## APPENDIX A   PSEUDOCODE FOR ADDITIONAL ALGORITHMS AND SUBROUTINES

**Pseudocode for sampling an expression from the RNN.** The sampling process in DSR (line 4 of Algorithm 1) is more complicated than typical autoregressive sampling procedures due to applying constraints in situ and providing hierarchical information to the RNN. Thus, we provide pseudocode for this process in Algorithm 2. Within this algorithm, the function $\text{Arity}(\tau_i)$ simply returns the arity (number of arguments) of token $\tau_i$, i.e. two for binary operators, one for unary operators, or zero for input variables or constants.

---

**Algorithm 2** Sampling an expression from the RNN

---

**input**  RNN with parameters $\theta$; library of tokens $\mathcal{L}$
**output**  Pre-order traversal $\tau$ of an expression sampled from the RNN

  1: $\tau \leftarrow []$                                                    ▷ Initialize empty traversal
  2: counter $\leftarrow 1$                         ▷ Initialize counter for number of unselected nodes
  3: $x \leftarrow \text{empty}\|\text{empty}$                  ▷ Initial RNN input is empty parent and sibling
  4: $c_0 \leftarrow \vec{0}$                                     ▷ Initialize RNN cell state to zero
  5: **for** $i = 1, 2, \ldots$ **do**
  6:     $(\psi^{(i)}, c_i) \leftarrow \text{RNN}(x, c_{i-1}; \theta)$           ▷ Emit probabilities; update state
  7:     $\psi^{(i)} \leftarrow \text{ApplyConstraints}(\psi^{(i)}, \mathcal{L}, \tau)$         ▷ Adjust probabilities
  8:     $\tau_i \leftarrow \text{Categorical}(\psi^{(i)})$               ▷ Sample next token
  9:     $\tau \leftarrow \tau \| \tau_i$                           ▷ Append token to traversal
10:     counter $\leftarrow$ counter $+ \text{Arity}(\tau_i) - 1$     ▷ Update number of unselected nodes
11:     **if** counter $= 0$ **then return** $\tau$         ▷ If expression is complete, return it
12:     $x \leftarrow \text{ParentSibling}(\tau)$            ▷ Compute next parent and sibling

---

**Additional subroutines.** DSR includes several subroutines used when sampling an expression from the RNN and during training. In Subroutine 1, we describe the function $\text{ParentSibling}(\tau)$ used in Algorithm 2, which computes the parent and sibling of the next token to be sampled. This subroutine uses the following logic. If the final node in the partial traversal is a unary or binary operator, then that node is the parent and there is no sibling. Otherwise, the subroutine iterates backward through the traversal until finding a node with an unselected child node. That node is the parent and the subsequent node is the sibling.

---

**Subroutine 1** Computing parent and sibling inputs to the RNN

---

**input**  Partially sampled traversal $\tau$
**output**  Concatenated parent and sibling tokens of the next token to be sampled

  1: $T \leftarrow |\tau|$                                        ▷ Length of partial traversal
  2: counter $\leftarrow 0$                     ▷ Initialize counter for number of unselected nodes
  3: **if** $\text{Arity}(\tau_T) > 0$ **then**
  4:     parent $\leftarrow \tau_T$
  5:     sibling $\leftarrow$ empty
  6:     **return** parent$\|$sibling
  7: **for** $i = T, \ldots, 1$ **do**                              ▷ Iterate backward
  8:     counter $\leftarrow$ counter $+ \text{Arity}(\tau_i) - 1$     ▷ Update number of unselected nodes
  9:     **if** counter $= 0$ **then**
10:        parent $\leftarrow \tau_i$
11:        sibling $\leftarrow \tau_{i+1}$
12:        **return** parent$\|$sibling

---

In Subroutine 2, we describe the function $\text{ApplyConstraints}(\psi, \mathcal{L}, \tau)$ used in Algorithm 2, which zeros out the probabilities of tokens that would violate any given constraints. Within this subroutine, the user-specific function $\text{ViolatesConstraint}(\tau, \mathcal{L}_i)$ returns TRUE if adding the $i^{\text{th}}$ token of the library $\mathcal{L}$ to the partial traversal $\tau$ would violate any user-specified constraints, and FALSE otherwise.

In Subroutine 3, we describe the function $\text{OptimizeConstants}(\tau, R)$ used in Algorithm 1, which optimizes the placeholder constants $\xi$ of an expression with respect to the reward function using a

---

**Subroutine 2** Applying generic constraints in situ when sampling from the RNN

---

**input** Categorical probabilities $\psi$; corresponding library of tokens $\mathcal{L}$; partially sampled traversal $\tau$
**output** Adjusted categorical probabilities $\psi$
  1: $L \leftarrow |\mathcal{L}|$                                                                    ▷ Length of library
  2: **for** $i = 1, \ldots, L$ **do**
  3:     **if** ViolatesConstraint$(\tau, \mathcal{L}_i)$ **then** $\psi_i \leftarrow 0$     ▷ If the token would violate a constraint, set its probability to 0
  4: $\psi \leftarrow \frac{\psi}{\sum_i \psi_i}$                                           ▷ Normalize probability vector back to 1
  5: **return** $\psi$

---

black-box optimizer, e.g. BFGS. In Algorithm 1, this subroutine is abstracted away into the computation of the reward function; here, we explicitly consider the reward function $R(\tau; \xi)$ as parameterized by the placeholder constants. Within this subroutine, the function ReplaceConstants$(\tau, \xi^\star)$ replaces the placeholder constants in the expression with the optimized constants $\xi^\star$.

---

**Subroutine 3** Optimizing the constants of an expression (inner optimization loop)

---

**input** Expression $\tau$ with placeholder constants $\xi$; reward function $R$
**output** Expression $\tau^\star$ with optimized constants $\xi^\star$
  1: $\xi^\star \leftarrow \arg\max_\xi R(\tau; \xi)$                  ▷ Maximize reward w.r.t. constants, e.g. with BFGS
  2: $\tau^\star \leftarrow$ ReplaceConstants$(\tau, \xi^\star)$                    ▷ Replace placeholder constants
  3: **return** $\tau^\star$

---

## APPENDIX B   PROOF OF POLICY GRADIENT FOR RISK-SEEKING OBJECTIVE (PROPOSITION 1)

Tamar et al. (2014) proved the following result for the policy gradient of the $\varepsilon$-conditional value at risk objective, $J_{\text{CVaR}}(\theta)$:

$$\nabla_\theta J_{\text{CVaR}}(\theta) \doteq \nabla_\theta \mathbb{E}_{\tau \sim p(\tau|\theta)} \left[ R(\tau) \mid R(\tau) \leq R_\varepsilon(\theta) \right]$$
$$= \mathbb{E}_{\tau \sim p(\tau|\theta)} \left[ (R(\tau) - R_\varepsilon(\theta)) \nabla_\theta \log p(\tau|\theta) \mid R(\tau) \leq R_\varepsilon(\theta) \right]$$

For completeness, we adapt this proof from the CVaR objective to our risk-seeking objective given in (1). We emphasize that the proof closely follows Tamar et al. (2014). The difference amounts to defining the threshold $R_\varepsilon$ as the *top* $\varepsilon$ quantile of rewards (instead of the bottom $\varepsilon$ quantile), and replacing $R(\tau) \leq R_\varepsilon$ in the CVaR objective with $R(\tau) \geq R_\varepsilon$ in our objective. This difference results in the limits of integration swapping, causing an additional minus sign that eventually cancels out. Important differences from the proof in Tamar et al. (2014) are colored in red. As in Tamar et al. (2014), we first demonstrate the policy gradient in the single variable case, then extend to the multivariable case in the reinforcement learning setting. The assumptions follow Assumptions 1 - 7 detailed in Tamar et al. (2014).

*Proof.* Consider a bounded random variable $Z \in [-b, b]$ generated from a parameterized distribution $p(Z|\theta)$. The $(1 - \varepsilon)$ quantile of $Z$ is:

$$Q_{1-\varepsilon}(Z; \theta) = \inf\{z : \text{CDF}(z) \geq 1 - \varepsilon\},$$

where $\text{CDF}(z)$ is the cumulative distribution function corresponding to $p(Z|\theta)$. We define the risk-seeking objective $J_{\text{risk}}(\theta; \varepsilon)$ as the expectation of the $\varepsilon$ fraction of the *best* outcomes of $Z$:

$$J_{\text{risk}}(\theta; \varepsilon) \doteq \mathbb{E}_{Z \sim p(Z|\theta)} \left[ Z \mid Z \geq Q_{1-\varepsilon}(Z; \theta) \right] \tag{2}$$

We define $D_\theta$ as the set of all values of $z$ *above* this quantile:

$$D_\theta = \{ z \in [-b, b] : z \geq Q_{1-\varepsilon}(Z; \theta) \}$$

By construction, $D_\theta$ is simply the interval $[Q_{1-\varepsilon}(Z; \theta), b]$, and

$$\int_{z \in D_\theta} p(z|\theta) dz = \varepsilon \tag{3}$$

Rewriting the conditional expectation in Equation (2) as an integral,

$$J_{\text{risk}}(\theta; \varepsilon) = \frac{1}{\int_{z \in D_\theta} p(z|\theta)dz} \int_{z \in D_\theta} p(z|\theta)zdz$$

$$= \frac{1}{\varepsilon} \int_{z \in D_\theta} p(z|\theta)zdz$$

$$= \frac{1}{\varepsilon} \int_{Q_{1-\varepsilon}(Z;\theta)}^{b} p(z|\theta)zdz$$

We now compute the gradient of $J_{\text{risk}}(\theta; \varepsilon)$ with respect to $\theta$. In the standard policy gradient derivation, the gradient can be swapped with the integral. In this case, the domain of integration depends on $\theta$, thus requiring the Leibniz rule:

$$\nabla_\theta J_{\text{risk}}(\theta; \varepsilon) = \nabla_\theta \frac{1}{\varepsilon} \int_{Q_{1-\varepsilon}(Z;\theta)}^{b} p(z|\theta)zdz$$

$$= \frac{1}{\varepsilon} \int_{Q_{1-\varepsilon}(Z;\theta)}^{b} \nabla_\theta p(z|\theta)zdz - \frac{1}{\varepsilon}p(Q_{1-\varepsilon}(Z;\theta)|\theta)Q_{1-\varepsilon}(Z;\theta)\nabla_\theta Q_{1-\varepsilon}(Z;\theta) \quad (4)$$

We similarly take the gradient of Equation (3):

$$0 = \nabla_\theta \int_{z \in D_\theta} p(z|\theta)dz$$

$$= \nabla_\theta \int_{Q_{1-\varepsilon}(Z;\theta)}^{b} p(z|\theta)dz$$

$$= \int_{Q_{1-\varepsilon}(Z;\theta)}^{b} \nabla_\theta p(z|\theta)dz - p(Q_{1-\varepsilon}(Z;\theta)|\theta)\nabla_\theta Q_{1-\varepsilon}(Z;\theta) \quad (5)$$

Plugging Equation (5) into Equation (4) and rearranging yields

$$\nabla_\theta J_{\text{risk}}(\theta; \varepsilon) = \frac{1}{\varepsilon} \int_{Q_{1-\varepsilon}(Z;\theta)}^{b} \nabla_\theta p(z|\theta) \left(z - Q_{1-\varepsilon}(Z;\theta)\right) dz$$

Using the "log-derivative trick" (multiplying by $p(Z|\theta)/p(Z|\theta)$ and using the derivative of a logarithm) and the definition of conditional expectation yields the final result:

$$\nabla_\theta J_{\text{risk}}(\theta; \varepsilon) = \frac{1}{\varepsilon} \int_{Q_{1-\varepsilon}(Z;\theta)}^{b} (z - Q_{1-\varepsilon}(Z;\theta)) \, p(z|\theta)\nabla_\theta \log p(z|\theta)dz$$

$$= \mathbb{E}_{Z \sim p(Z|\theta)} \left[(Z - Q_{1-\varepsilon}(Z;\theta))\nabla_\theta \log p(Z|\theta) \mid Z \geq Q_{1-\varepsilon}(Z;\theta)\right]$$

The extension to the case in which $Z$ is replaced by a scalar reward function $R(\tau)$, where $\tau$ is generated from a parameterized distribution $p(\tau|\theta)$, follows the proof in Tamar et al. (2014) without additional adaptation. Thus,

$$\nabla_\theta J_{\text{risk}}(\theta; \varepsilon) = \mathbb{E}_{\tau \sim p(\tau|\theta)} \left[(R(\tau) - R_\varepsilon(\theta)) \nabla_\theta \log p(\tau|\theta) \mid R(\tau) \geq R_\varepsilon(\theta)\right],$$

where $R_\varepsilon(\theta)$ is the $(1 - \varepsilon)$-quantile of the distribution of rewards under the current policy. $\qquad \square$

## APPENDIX C    DESCRIPTIONS OF BASELINE ALGORITHMS

**Priority queue training.** The PQT baseline was implemented using our DSR framework (including in situ constraints, hierarchical RNN inputs, and constant optimization), but replacing the risk-seeking policy gradient objective with the priority queue training objective proposed in Abolafia et al. (2018). PQT works by maintaining a priority queue, defined by the top $k$ highest reward samples ever encountered during training, which is updated each training step. They then define their training objective as the average log-likelihood of the samples in the priority queue:

$$J_{\text{PQT}}(\theta; k) \doteq \frac{1}{k} \sum_{i=1}^{k} \log p(\tau^{[i]}|\theta),$$

where $k$ is a hyperparameter controlling the size of the priority queue, and $\tau^{[i]}$ is the $i^{\text{th}}$ member of the priority queue, i.e. the $i^{\text{th}}$ best expression encountered during training. Unlike $J_{\text{std}}(\theta)$ and $J_{\text{risk}}(\theta; \varepsilon)$, the terms in $J_{\text{PQT}}(\theta; k)$ are not scaled by rewards.

As discussed in Abolafia et al. (2018), the PQT objective can be combined with a policy gradient objective; however, the authors did not find this to improve results relative to PQT only. For this reason, and to avoid additional hyperparameters, our PQT baseline does not include a policy gradient term.

**Vanilla policy gradient.** The VPG baseline was implemented using our DSR framework (including in situ constraints, hierarchical RNN inputs, and constant optimization), but replacing the risk-seeking policy gradient objective, $J_{\text{risk}}(\theta; \varepsilon)$, with the standard policy gradient objective, $J_{\text{std}}(\theta)$. In addition, as is common for standard policy gradients, we subtract a baseline from the reward function. As in Zoph & Le (2017), our baseline is an exponentially-weighted moving average (EWMA) of rewards. This introduces a VPG-specific hyperparameter $\beta$, controlling the degree of weighting decrease. Note that VPG is essentially an ablation of the risk-seeking policy gradient in DSR. However, as it is the critical ablation, we independently tune hyperparameters for the VPG baseline, in contrast to the "No risk" ablation in Figure 3.

**Genetic programming.** GP-based symbolic regression was implemented using the open-source software package "deap" (Fortin et al., 2012). The initial population of expressions is generated using the "full" method (Koza, 1992) with depth randomly selected between $d_{\min}$ and $d_{\max}$. The selection operator is defined by deterministic tournament selection, in which the expression with the best fitness among $k$ randomly selected expressions is chosen. The crossover operator is defined by swapping random subtrees between two expressions. The point mutation operator is defined by replacing a random subtree with a new subtree initialized using the "full" method with depth randomly selected between $d_{\min}$ and $d_{\max}$.

Constraints were implemented in GP in a post hoc manner. Each time a new expression is generated via mutation or crossover, all constraints are checked. If any constraints are violated, the new expression is rejected and the evolutionary operation is simply reverted (Fortin et al., 2012; Craenen et al., 2001). Like the in situ constraints used in DSR, this process reduces the search space; however, unlike in situ constraints, this rejection-based process introduces a tradeoff between search space and sample diversity. Thus, we performed all GP hyperparameter grid search both with and without post hoc constraints (using the same constraints as DSR and the RNN-based baselines). Using the best hyperparameters found for each mode, we repeated the full recovery rate experiments on the Nguyen benchmarks, and ultimately selected the best performing mode. We found that constraints actually hindered performance: GP recovery rate with constraints fell from $60.1\%$ (in Table 1) to $32.3\%$ across the Nguyen benchmarks. Thus, GP results herein do not include constraints other than constraining the maximum length to 30 (to prevent excessively long expressions with a low chance of recovery) and the maximum number of constants to 3 (otherwise, runs using constant optimization can become prohibitively expensive) as in DSR.

**Eureqa.** Considered to be the gold standard for symbolic regression (Udrescu & Tegmark, 2020), Eureqa is based on the GP-based approach proposed by Schmidt & Lipson (2009). Eureqa experiments were run using the DataRobot platform (www.datarobot.com). Eureqa allows the user to select which tokens to include, and thus experiments used the same library as DSR, as specified by Table 2 in Appendix D. While the number of expression evaluations is not tunable or made transparent to the user, Eureqa allows the user to select among several runtime modes. For all experiments, we used the longest-running mode, `Long Running Search (10,000 Generations)`. After training, Eureqa outputs the best found expression based on mean-square error (MSE), as well as the Pareto front (using a proprietary complexity measure). For noise experiments, to mimic the other baselines, we checked for exact symbolic equivalence for all solutions along the Pareto front.

**Wolfram.** Wolfram results were obtained using the Mathematica built-in function `FindFormula`, which combines Markov chain Monte Carlo sampling methods with nonlinear regression (Fortuna, 2015). The function allows the user to specify the set of tokens to include. However, subtraction and division operators are not allowed. The use of real-valued constants is an integral part of the method and cannot be deactivated. Therefore, the Wolfram experiments used a slightly different library than the other algorithms. Specifically, we used $\{+, \times, \text{pow}, \sin, \cos, \exp, \log, x\}$, where $\text{pow}(a, b) \doteq a^b$.

Note that $a - b$ can be obtained via $a + (-1.0) \times b$ and $a \div b$ can be obtained via $a \times \text{pow}(b, -1.0)$; thus, all benchmarks can still be recovered using this modified library.

**Notable exclusions.** The baselines selected for this work are suitable for searching the space of tractable mathematical expressions to discover exact symbolic expressions, and can accommodate numerical constants within expressions. There are several well-known algorithms that do not exhibit these features, and thus we do not include as baselines:

- Geometric-semantic genetic programming (GSGP): Originally proposed by Moraglio et al. (2012), GSGP is a GP-based approach that directly searches the space of the *semantics* of a program. They do this by constructing geometric semantic operators, which promote a strict increase in fitness per generation. However, this desirable feature comes with the cost of the size of the expressions growing exponentially. The resulting expressions are routinely on the order of *trillions* of nodes (Pawlak, 2016). This essentially forfeits the ability to exactly recover expressions, and severely limits the usefulness of the technique for interpretability—the primary motivation for symbolic regression.

- Equation learner (EQL$^{\div}$): As discussed in Related Work, Sahoo et al. (2018) perform symbolic regression via neural networks whose activation functions are elementary operators. However, the approach precludes the ability to recovery many simple classes of expressions, such as those involving exponent, logarithm, roots, or division within unary operators. Further, the resulting expressions can be very long, including real-valued shifts and scaled for each operator.

- AI Feynman (AIF): As discussed in Related Work, Udrescu & Tegmark (2020) develop a tool to recursively simplify symbolic regression problems into smaller sub-problems; solving each sub-problem then requires an inner search algorithm. The authors demonstrate that many physics equations can be reduced to sub-problems that are solvable using a simple inner search algorithm, e.g. polynomial fit or small brute-force search. However, more challenging sub-problems—including many of the benchmark expressions in this work (namely, those with numerical constants, or those which are non-separable)—still require an underlying symbolic regression method to conduct the discrete search. Thus, AIF can be used during pre-processing as a problem-simplification step, then combined with *any* symbolic regression algorithm. Here, DSR and the baselines tested focus on the underlying discrete search problem, e.g. after any problem simplification steps have been applied.

## APPENDIX D    ADDITIONAL EXPERIMENT DETAILS

**Detailed description of the symbolic regression benchmarks.** Details of the benchmark symbolic regression problems are shown in Table 2. Note that benchmarks without the "const" token in the library do note use a constant optimizer (line 4 in Algorithm 1, detailed in 3 in Appendix A). Benchmarks without real-valued constants can be recovered exactly (except Neat-6, discussed in Appendix E), thus recovery is defined by exact symbolic equivalence. Note that the square root and power operators are not part of the function set; however, such benchmarks can still be recovered with the given function set, e.g. Nguyen-8 can still be recovered via $\exp(\frac{x}{x+x}\log(x))$ and Nguyen-11 can still be recovered via $\exp(y\log(x))$.

While all algorithms always produce *syntactically* valid expressions, they do not always produce *semantically* meaningful expressions. For example, the expression $\log(x)$ is not semantically meaningful for non-positive values of $x$. Thus, for all non-commercial algorithms, any expression that produces an overflow or other floating-point error on the input domain is deemed "invalid" and receives a reward of 0, which corresponds to infinite error. Eureqa and Wolfram follow their own strategy to deal with this issue.

**Hyperparameter selection.** Hyperparameters were tuned by performing grid search on benchmarks Nguyen-7 and Nguyen-10. For each hyperparameter combination, we performed 10 independent training runs of the algorithm for 1M total expression evaluations. We selected the hyperparameter combination with the highest average recovery rate, with ties broken by lowest average NRMSE. For all algorithms, the best found hyperparameters were used for all experiments and all benchmark expressions.

Table 2: Benchmark symbolic regression problem specifications. Input variables are denoted by $x$ and/or $y$. $U(a,b,c)$ denotes $c$ random points uniformly sampled between $a$ and $b$ for each input variable; training and test datasets use different random seeds. $E(a,b,c)$ denotes $c$ points evenly spaced between $a$ and $b$ for each input variable; training and test datasets use the same points (except Neat-6, which uses $E(1,120,120)$ as test data, and the Jin tests, which use $U(-3,3,30)$ as test data). To simplify notation, libraries are defined relative to a "base" library $\mathcal{L}_0 = \{+, -, \times, \div, \sin, \cos, \exp, \log, x\}$. Placeholder operands are denoted by $\bullet$, e.g. $\bullet^2$ corresponds to the square operator.

| Name | Expression | Dataset | Library |
|---|---|---|---|
| Nguyen-1 | $x^3 + x^2 + x$ | $U(-1,1,20)$ | $\mathcal{L}_0$ |
| Nguyen-2 | $x^4 + x^3 + x^2 + x$ | $U(-1,1,20)$ | $\mathcal{L}_0$ |
| Nguyen-3 | $x^5 + x^4 + x^3 + x^2 + x$ | $U(-1,1,20)$ | $\mathcal{L}_0$ |
| Nguyen-4 | $x^6 + x^5 + x^4 + x^3 + x^2 + x$ | $U(-1,1,20)$ | $\mathcal{L}_0$ |
| Nguyen-5 | $\sin(x^2)\cos(x) - 1$ | $U(-1,1,20)$ | $\mathcal{L}_0$ |
| Nguyen-6 | $\sin(x) + \sin(x + x^2)$ | $U(-1,1,20)$ | $\mathcal{L}_0$ |
| Nguyen-7 | $\log(x+1) + \log(x^2+1)$ | $U(0,2,20)$ | $\mathcal{L}_0$ |
| Nguyen-8 | $\sqrt{x}$ | $U(0,4,20)$ | $\mathcal{L}_0$ |
| Nguyen-9 | $\sin(x) + \sin(y^2)$ | $U(0,1,20)$ | $\mathcal{L}_0 \cup \{y\}$ |
| Nguyen-10 | $2\sin(x)\cos(y)$ | $U(0,1,20)$ | $\mathcal{L}_0 \cup \{y\}$ |
| Nguyen-11 | $x^y$ | $U(0,1,20)$ | $\mathcal{L}_0 \cup \{y\}$ |
| Nguyen-12 | $x^4 - x^3 + \frac{1}{2}y^2 - y$ | $U(0,1,20)$ | $\mathcal{L}_0 \cup \{y\}$ |
| Nguyen-2$'$ | $4x^4 + 3x^3 + 2x^2 + x$ | $U(-1,1,20)$ | $\mathcal{L}_0$ |
| Nguyen-5$'$ | $\sin(x^2)\cos(x) - 2$ | $U(-1,1,20)$ | $\mathcal{L}_0$ |
| Nguyen-8$'$ | $\sqrt[3]{x}$ | $U(0,4,20)$ | $\mathcal{L}_0$ |
| Nguyen-8$''$ | $\sqrt[3]{x^2}$ | $U(0,4,20)$ | $\mathcal{L}_0$ |
| Nguyen-1$^c$ | $3.39x^3 + 2.12x^2 + 1.78x$ | $U(-1,1,20)$ | $\mathcal{L}_0 \cup \{\text{const}\}$ |
| Nguyen-5$^c$ | $\sin(x^2)\cos(x) - 0.75$ | $U(-1,1,20)$ | $\mathcal{L}_0 \cup \{\text{const}\}$ |
| Nguyen-7$^c$ | $\log(x+1.4) + \log(x^2+1.3)$ | $U(0,2,20)$ | $\mathcal{L}_0 \cup \{\text{const}\}$ |
| Nguyen-8$^c$ | $\sqrt{1.23x}$ | $U(0,4,20)$ | $\mathcal{L}_0 \cup \{\text{const}\}$ |
| Nguyen-10$^c$ | $\sin(1.5x)\cos(0.5y)$ | $U(0,1,20)$ | $\mathcal{L}_0 \cup \{y, \text{const}\}$ |
| GrammarVAE-1 | $\frac{1}{3} + x + \sin(x^2)$ | $E(-10,10,10^3)$ | $\mathcal{L}_0 - \{-, \cos, \log\} \cup \{1,2,3\}$ |
| Jin-1 | $2.5x^4 - 1.3x^3 + 0.5y^2 - 1.7y$ | $U(-3,3,100)$ | $\mathcal{L}_0 - \{\log\} \cup \{\bullet^2, \bullet^3, y, \text{const}\}$ |
| Jin-2 | $8.0x^2 + 8.0y^3 - 15.0$ | $U(-3,3,100)$ | $\mathcal{L}_0 - \{\log\} \cup \{\bullet^2, \bullet^3, y, \text{const}\}$ |
| Jin-3 | $0.2x^3 + 0.5y^3 - 1.2y - 0.5x$ | $U(-3,3,100)$ | $\mathcal{L}_0 - \{\log\} \cup \{\bullet^2, \bullet^3, y, \text{const}\}$ |
| Jin-4 | $1.5\exp(x) + 5.0\cos(y)$ | $U(-3,3,100)$ | $\mathcal{L}_0 - \{\log\} \cup \{\bullet^2, \bullet^3, y, \text{const}\}$ |
| Jin-5 | $6.0\sin(x)\cos(y)$ | $U(-3,3,100)$ | $\mathcal{L}_0 - \{\log\} \cup \{\bullet^2, \bullet^3, y, \text{const}\}$ |
| Jin-6 | $1.35xy + 5.5\sin((x-1.0)(y-1.0))$ | $U(-3,3,100)$ | $\mathcal{L}_0 - \{\log\} \cup \{\bullet^2, \bullet^3, y, \text{const}\}$ |
| Neat-1 | $x^4 + x^3 + x^2 + x$ | $U(-1,1,20)$ | $\mathcal{L}_0 \cup \{1\}$ |
| Neat-2 | $x^5 + x^4 + x^3 + x^2 + x$ | $U(-1,1,20)$ | $\mathcal{L}_0 \cup \{1\}$ |
| Neat-3 | $\sin(x^2)\cos(x) - 1$ | $U(-1,1,20)$ | $\mathcal{L}_0 \cup \{1\}$ |
| Neat-4 | $\log(x+1) + \log(x^2+1)$ | $U(0,2,20)$ | $\mathcal{L}_0 \cup \{1\}$ |
| Neat-5 | $2\sin(x)\cos(y)$ | $U(-1,1,100)$ | $\mathcal{L}_0 \cup \{y\}$ |
| Neat-6 | $\sum_{k=1}^{x} \frac{1}{k}$ | $E(1,50,50)$ | $\{+, \times, \div, \bullet^{-1}, -\bullet, \sqrt{\bullet}, x\}$ |
| Neat-7 | $2 - 2.1\cos(9.8x)\sin(1.3y)$ | $E(-50,50,10^5)$ | $\mathcal{L}_0 \cup \{\tan, \tanh, \bullet^2, \bullet^3, \sqrt{\bullet}, y\}$ |
| Neat-8 | $\frac{e^{-(x-1)^2}}{1.2 + (y-2.5)^2}$ | $U(0.3,4,100)$ | $\{+, -, \times, \div, \exp, e^{-\bullet}, \bullet^2, x, y\}$ |
| Neat-9 | $\frac{1}{1+x^{-4}} + \frac{1}{1+y^{-4}}$ | $E(-5,5,21)$ | $\mathcal{L}_0 \cup \{y\}$ |

Conveniently, the three RNN-based algorithms (DSR, PQT, and VPG) each have one unique hyperparameter: risk factor $\varepsilon$ for DSR, priority queue size $k$ for PQT, and EWMA coefficient $\beta$ for VPG. The remaining three hyperparameters considered (batch size, learning rate, and entropy weight) are present for each algorithm. For these three algorithms, the space of hyperparameters considered was batch size $\in \{250, 500, 1000\}$, learning rate $\in \{0.0003, 0.0005, 0.001\}$, and entropy weight $\lambda_{\mathcal{H}} \in \{0.01, 0.05, 0.1\}$. For the algorithm-specific hyperparameters, we considered risk factor $\varepsilon \in \{0.05, 0.10, 0.15\}$ for DSR, priority queue size $k \in \{5, 10, 20\}$ for PQT, and EWMA coefficient $\beta \in \{0.1, 0.25, 0.5\}$ for VPG. Thus, these algorithms each considered 81 hyperparameter combinations. The final tuned hyperparameters are listed in Table 3. (Note that hyperparameters were tuned independently for each algorithm; identical values across algorithms are incidental.)

For GP, the space of hyperparameters considered was population size $\in \{100, 250, 500, 1000\}$, tournament size $\in \{2, 3, 5, 10\}$, mutation probability $\in \{0.01, 0.03, 0.05, 0.10, 0.15\}$, crossover probability $\in \{0.25, 0.50, 0.75, 0.90, 0.95\}$, and post hoc constraints $\in \{\textsc{True}, \textsc{False}\}$ (800 combinations). The final tuned hyperparameters are listed in Table 4.

As commercial software, Eureqa and Wolfram do not expose tunable hyperparameters. Wolfram allows the user to select among three "performance goal" modes (`Error`, `Score`, and `Complexity`)

and two "specificity goal" modes (Low and High). We tested all 6 combinations for 100 random seeds. Wolfram is not applicable to Nguyen-10, and we found that recovery rate for Nguyen-7 was 0% for all combinations; thus, the best combination of modalities was selected based on benchmarks Nguyen-5 and Nguyen-6 instead. The best results were obtained using Error for performance and High for specificity.

Table 3: Tuned hyperparameters for RNN-based algorithms.

| Parameter name | Symbol | DSR | PQT | VPG |
|---|---|---|---|---|
| Batch size | $N$ | 1000 | 1000 | 1000 |
| Learning rate | $\alpha$ | 0.0005 | 0.0005 | 0.0001 |
| Entropy coefficient | $\lambda_{\mathcal{H}}$ | 0.005 | 0.005 | 0.005 |
| Risk factor | $\varepsilon$ | 0.05 | – | – |
| Priority queue size | $k$ | – | 10 | – |
| EWMA coefficient | $\alpha$ | – | – | 0.25 |

Table 4: Tuned hyperparameters for GP.

| Parameter | Value |
|---|---|
| Population size | 1,000 |
| Fitness function | NRMSE |
| Initialization method | Full |
| Selection type | Tournament |
| Tournament size ($k$) | 2 |
| Mutation probability | 0.05 |
| Crossover probability | 0.95 |
| Post hoc constraints | FALSE |
| Minimum subtree depth ($d_{\min}$) | 0 |
| Maximum subtree depth ($d_{\max}$) | 2 |

**Additional details for constant optimization.** The same constant optimizer was used for all non-commercial software algorithms: DSR, PQT, VPG, and GP. To optimize constants, the values for constant placeholder tokens are optimized against the reward (or fitness) function using BFGS with an initial guess of 1.0 for each constant. Before training, we ensured that all benchmarks with constants do not get stuck in a poor local optimum when optimizing with BFGS and the candidate functional form is correct. Since numerical constants can only be recovered up to floating-point precision, for benchmarks with constants we determined recovery by manually inspecting the functional form for symbolic correctness. Since constant optimization is a computational bottleneck, we limited each expression to three constants for all experiments with the "const" token, and ran experiments for 1M expression evaluations instead of 2M. Eureqa and Wolfram follow their own strategy to learn real-valued constants.

**Additional details for ablation studies.** In Figure 3, "No risk" denotes using the standard policy gradient (with baseline) instead of the risk-seeking policy gradient, equivalent to $\varepsilon = 1$. Since the standard policy gradient typically includes a baseline term, we used an exponentially-weighted moving average (weight 0.5) of the average reward of the batch. The difference between this ablation and the VPG baseline is that VPG hyperparameters were tuned independently, whereas this ablation uses the DSR tuned hyperparameters. "No entropy" denotes no entropy bonus, equivalent to $\lambda_{\mathcal{H}} = 0$. "No constraints" denotes no constraints precluding nested trigonometric operators, inverse unary operators, minimum length, or maximum length. Instead, if the maximum length of 30 tokens is reached, the expression is appended with $x$ until complete. "No risk/ent." denotes combining ablations for "No risk" and "No entropy." "No risk/con." denotes combining ablations for "No risk" and "No constraints." "No ent./con." denotes combining ablations for "No entropy" and "No constraints." "All ablations" denotes combining ablations for "No risk," "No entropy," and "No constraints."

**Pareto front computation.** Overfitting in symbolic regression occurs when an expression has added complexity to conform to noisy data. For example, high-degree polynomials can fit small datasets with low error, though generalization may be poor. While overfitting is not an issue for noiseless experiments (since the global optimal expression has zero error), it is indeed possible for expressions to overfit in the noisy data experiments, especially with small dataset sizes. Thus, for noisy data

experiments, we defined recovery as exact symbolic equivalence on any expression along the Pareto front (defined over the reward-complexity plane) at the end of training. To compute the Pareto front, we define a simple complexity measure $C$:

$$C(\tau) = \sum_{i}^{T} c(\tau_i),$$

where $c$ is the complexity of a particular token. In accordance with Eureqa's default parameters, we used token complexities of 1 for $+, -, \times$, input variables, and constants; 2 for $\div$; 3 for $\sin$ and $\cos$; and 4 for $\exp$ and $\log$.

Note that $C$ is only used to determine the Pareto front at the end of training; it does not affect training. This method of determining recovery based on the Pareto front was used for all algorithms, except that Eureqa uses a proprietary complexity measure.

**Computing infrastructure.** Experiments were executed on an Intel Xeon E5-2695 v4 equipped with NVIDIA Tesla P100 GPUs, with 32 cores per node, 2 GPUs per node, and 256 GB RAM per node.

## APPENDIX E    COMPARISONS TO LITERATURE-REPORTED RESULTS

Unfortunately, the majority of symbolic regression methods are inconsistent in their experimental setup, i.e. choice of library, benchmarks, and measure of performance. Often, benchmark expressions are hand-selected from several benchmark suites (Trujillo et al., 2016) or even hand-crafted for a particular study (Kusner et al., 2017; Sahoo et al., 2018). The criteria for determining "recovery" (also called "hit" or "success") is often defined by an arbitrary error threshold instead of exact symbolic equivalence (White et al., 2013). Further, different works use different performance metrics (e.g. RMSE, mean absolute error) that cannot be converted without access to raw data (Jin et al., 2019).

These issues make it difficult to compare methods via direct comparison of literature-reported results. To enable such comparisons, we mimic the experimental setup and measures of performance for several works. The alternative approach would be to evaluate these methods using our experimental setup and measures of performance; however, not all methods have available source code or sufficient implementation details for reproducibility. Below, we recapitulate the experiments of several additional works, and directly compare literature-reported results to our experiments using DSR. We note that even these comparisons are imperfect for for benchmarks with randomly sampled points, as the random number generator and seed used to generate the training and test data are typically not disclosed.

**Semantics-based symbolic regression.** Huynh et al. (2016) is an exceptional case in that they also report recovery as exact symbolic recovery, using an overlapping set of benchmarks: Nguyen-1 through Nguyen-10. In this work, the authors present Semantic-based Symbolic Regression (SSR), a GP-based approach that considers the *semantics* of each expression tree; for example, they remove duplicate trees with different tree structures but identical semantics, e.g. $((((x-1)+(x/x))-x)+y)$ is equivalent to $y$. In Table 5, we compare DSR's recovery to SSR's reported values. To replicate their experimental setup, we report DSR recovery rate for 30 independent training runs. DSR's average recovery rate is more than twice as high as SSR.

Table 5: Recovery rate comparison of DSR and literature-reported values from SSR (Huynh et al., 2016). A bold value represents statistical significance ($p < 10^{-3}$) across all benchmarks.

| Benchmark | Expression | DSR | SSR |
|-----------|------------|-----|-----|
| Nguyen-1 | $x^3 + x^2 + x$ | 100% | 100% |
| Nguyen-2 | $x^4 + x^3 + x^2 + x$ | 100% | 10% |
| Nguyen-3 | $x^5 + x^4 + x^3 + x^2 + x$ | 100% | 70% |
| Nguyen-4 | $x^6 + x^5 + x^4 + x^3 + x^2 + x$ | 100% | 30% |
| Nguyen-5 | $\sin(x^2)\cos(x) - 1$ | 77% | 10% |
| Nguyen-6 | $\sin(x) + \sin(x + x^2)$ | 100% | 73% |
| Nguyen-7 | $\log(x + 1) + \log(x^2 + 1)$ | 33% | 3% |
| Nguyen-8 | $\sqrt{x}$ | 97% | 3% |
| Nguyen-9 | $\sin(x) + \sin(y^2)$ | 100% | 57% |
| Nguyen-10 | $2\sin(x)\cos(y)$ | 100% | 93% |
| | Average | **90.7%** | 45.0% |

**GrammarVAE.** As mentioned in Related Work, Kusner et al. (2017) develop a generative model for discrete objects that adhere to a pre-specified grammar, then optimize them in latent space. To mimic their experimental setup, we run 10 independent trials using the performance metric $\log(1 + \text{MSE})$ and the GrammarVAE benchmark described in Table 2. Using the best solution found in each trial, DSR's average performance metric is $0.0105 \pm 0.0149$, whereas GrammarVAE's reported average is orders of magnitude larger: $3.47 \pm 0.24$. In Table 6, as in Kusner et al. (2017), we compare the top 3 expressions reported in GrammarVAE to the top 3 expressions found using DSR across all 10 trials. In addition to exactly recovering the expression, DSR's top 3 expressions are all superior to those found by GrammarVAE. Further, when repeating for 100 trials, we find that DSR exactly recovers the benchmark expression in 19% of runs, whereas GrammarVAE never exactly recovers the ground truth expression.

Table 6: Comparison of DSR and literature-reported values from GrammarVAE (Kusner et al., 2017).

| Algorithm | Rank | Expression | $\log(1 + \text{MSE})$ |
|---|---|---|---|
| GrammarVAE | 1 | $\sin(3) + x + \sin(x^2)$ | 0.04 |
| GrammarVAE | 2 | $\frac{1}{2} + x + \sin(x^2)$ | 0.10 |
| GrammarVAE | 3 | $1 + x + \sin(x^2)$ | 0.37 |
| DSR | 1 | $\frac{1}{3} + x + \sin(x^2)$ | **0** |
| DSR | 2 | $\frac{\sin(1.5)}{3} + x + \sin(x^2)$ | $\mathbf{7.0 \times 10^{-7}}$ |
| DSR | 3 | $\sin\left(\frac{1}{3}\right) + x + \sin(x^2)$ | $\mathbf{3.8 \times 10^{-5}}$ |

**Bayesian Symbolic Regression.** In Jin et al. (2019), the authors propose Bayesian symbolic regression (BSR), a Bayesian framework for symbolic regression. In BSR, several carefully designed prior distributions are used to encode domain knowledge like preference of basis functions or tree structure. The resulting posterior distributions are efficiently sampled using Markov chain Monte Carlo techniques. To mimic their experiments, we run 50 independent trials using the performance metric RMSE and the Jin benchmarks described in Table 2. Table 7 shows the average RMSE on the test data for DSR and BSR across the benchmarks used in Jin et al. (2019). DSR outperforms BSR on all six benchmarks.

Table 7: RMSE comparison of DSR and literature-reported values from BSR (Jin et al., 2019).

| Benchmark | Expression | DSR | BSR |
|---|---|---|---|
| Jin-1 | $2.5x^4 - 1.3x^3 + 0.5y^2 - 1.7y$ | $\mathbf{0.46 \pm 0.41}$ | $2.04 \pm 3.27$ |
| Jin-2 | $8.0x^2 + 8.0y^3 - 15.0$ | $\mathbf{0 \pm 0}$ | $6.84 \pm 10.10$ |
| Jin-3 | $0.2x^3 + 0.5y^3 - 1.2y - 0.5x$ | $\mathbf{5.2 \times 10^{-4} \pm 3.5 \times 10^{-3}}$ | $0.21 \pm 0.20$ |
| Jin-4 | $1.5 \exp(x) + 5.0 \cos(y)$ | $\mathbf{1.36 \times 10^{-4} \pm 8.91 \times 10^{-4}}$ | $0.16 \pm 0.62$ |
| Jin-5 | $6.0 \sin(x) \cos(y)$ | $\mathbf{0 \pm 0}$ | $0.66 \pm 1.13$ |
| Jin-6 | $1.35xy + 5.5 \sin((x - 1.0)(y - 1.0))$ | $\mathbf{2.23 \pm 0.94}$ | $4.63 \pm 0.62$ |
| | Average | $\mathbf{0.45 \pm 0.23}$ | $2.42 \pm 2.66$ |

**Neat-GP.** In Trujillo et al. (2016), the authors propose Neat-GP, a genetic programming approach that uses the NEAT (NeuroEvolution of Augmenting Topologies) algorithm to address the problem of uncontrolled program size growth in standard GP approaches. The authors report median values of RMSE on the test data for 30 independent trials using the Neat benchmark set described in Table 2. We replicate their experimental setup using DSR and compare to their reported values in Table 8. DSR outperforms Neat-GP on seven of the nine benchmarks.

The task in Neat-6 is to find closed-form approximations of the harmonic series $H_n = \sum_{k=1}^{n} \frac{1}{k}$. As an additional experiment, we repeated DSR on Neat-6 with $\{\log, \text{const}\}$ added to $\mathcal{L}$. DSR discovered expressions which are remarkably accurate approximations of $H_n$: when extrapolating to all $n \in \mathbb{N}$, the error of the best discovered expression is less than $0.000001\%$. In particular, DSR discovered the expression:

$$H_n \approx \gamma + \log(n) + \frac{1}{2n + \frac{1}{11.3776 \frac{1}{n+15.725} + 0.327981}},$$

where $\gamma \approx 0.57721$ is the Euler-Mascheroni constant. This is a variation of the formula $H_n \approx \gamma + \log(n) + 1/(2n) - 1/(12n^2)$ found by Leonhard Euler in 1755 (Bromwich, 1908). Remarkably, the constant $\gamma$ naturally emerged in our recovered expression.

Table 8: Comparison of median values of RMSE for DSR and literature-reported values from Neat-GP (Trujillo et al., 2016).

| Benchmark | Expression | DSR | Neat-GP |
|-----------|------------|-----|---------|
| Neat-1 | $x^4 + x^3 + x^2 + x$ | **0** | 0.0779 |
| Neat-2 | $x^5 + x^4 + x^3 + x^2 + x$ | **0** | 0.0576 |
| Neat-3 | $\sin(x^2)\cos(x) - 1$ | **0.0041** | 0.0065 |
| Neat-4 | $\log(x+1) + \log(x^2+1)$ | **0.0189** | 0.0253 |
| Neat-5 | $2\sin(x)\cos(y)$ | **0** | 0.0023 |
| Neat-6 | $\sum_{k=1}^{x} \frac{1}{k}$ | **0.2378** | 0.2855 |
| Neat-7 | $2 - 2.1\cos(9.8x)\sin(1.3y)$ | 1.0606 | **1.0541** |
| Neat-8 | $\frac{e^{-(x-1)^2}}{1.2+(y-2.5)^2}$ | **0.1076** | 0.1498 |
| Neat-9 | $\frac{1}{1+x^{-4}} + \frac{1}{1+y^{-4}}$ | 0.1511 | **0.1202** |
| | Average | **0.1756** | 0.1977 |

## APPENDIX F ADDITIONAL RESULTS ON NGUYEN BENCHMARKS

**Variations of Nguyen benchmarks.** We introduce two sets of variations of the Nguyen benchmarks. In the first set, we alter existing benchmarks to make them more challenging. In the second set, we introduce real-valued constants and add the "const" token to the library to demonstrate the constant optimizer. We report recovery rate for DSR and all five baselines in Table 9. DSR significantly outperforms all baselines in the first set, and achieves 100% recover in the second set.

Table 9: Recovery rate comparison of DSR and five baselines on variations of the Nguyen symbolic regression benchmark suite. A bold value represents statistical significance ($p < 10^{-3}$) across each set of benchmarks.

| Benchmark | Expression | DSR | PQT | VPG | GP | Eureqa | Wolfram |
|-----------|------------|-----|-----|-----|-----|--------|---------|
| Nguyen-2' | $4x^4 + 3x^3 + 2x^2 + x$ | 96% | 92% | 0% | 52% | 0% | 100% |
| Nguyen-5' | $\sin(x^2)\cos(x) - 2$ | 87% | 25% | 0% | 4% | 1% | 0% |
| Nguyen-8' | $\sqrt[3]{x}$ | 50% | 44% | 1% | 2% | 0% | 0% |
| Nguyen-8'' | $\sqrt[3]{x^2}$ | 3% | 1% | 0% | 1% | 1% | 0% |
| | Average | **59.0%** | 40.5% | 0.2% | 14.8% | 0.5% | 25.0% |
| | | | | | | | |
| Nguyen-1$^c$ | $3.39x^3 + 2.12x^2 + 1.78x$ | 100% | 100% | 100% | 100% | 100% | 100% |
| Nguyen-5$^c$ | $\sin(x^2)\cos(x) - 0.75$ | 100% | 100% | 100% | 70% | 100% | 1% |
| Nguyen-7$^c$ | $\log(x+1.4) + \log(x^2+1.3)$ | 100% | 90% | 100% | 0% | 0% | 0% |
| Nguyen-8$^c$ | $\sqrt{1.23x}$ | 100% | 100% | 100% | 20% | 30% | 44% |
| Nguyen-10$^c$ | $\sin(1.5x)\cos(0.5y)$ | 100% | 100% | 40% | 0% | 10% | – |
| | Average | 100.0% | 98.0% | 88.0% | 38.0% | 48.0% | – |

**NRMSE comparisons.** Tables 1 and 9 show recovery rates for all algorithms across all benchmarks. In Table 10, we show the analogous results for NRMSE on the test data. DSR outperforms all algorithms across each benchmark set, except Eureqa on the original Nguyen benchmark set. Here, Eureqa's lower average NRMSE is attributed to its low error for Nguyen-12. However, this is a result of not being able to constrain or limit the complexity of expressions produced by Eureqa. In particular, Eureqa routinely identified expressions of length $\sim$100 for Nguyen-12, whereas all non-commercial baselines were limited to length 30 to ensure that expressions are tractable.

We also observe that for the few expressions with low or zero recovery rate (e.g. Nguyen-7 and Nguyen-12), GP sometimes exhibits lower NRMSE. One explanation is that GP is more prone to overfitting the expression to the dataset. As an evolutionary approach, GP directly modifies the previous generation's expressions, allowing it to make small "corrections" that decrease error each generation even if the functional form is far from correct. In contrast, in DSR the RNN "rewrites" each expression from scratch each iteration after learning from a gradient update, making it less prone to overfitting.

**Training curves.** Figures 6 and 7 show the reward ($1/(1+\text{NRMSE})$) and recovery rate, respectively, as a function of total expressions evaluated during training. Commercial software algorithms (Eureqa

Table 10: Comparison of NRMSE on the test data for DSR and five baselines on original and variations of the Nguyen symbolic regression benchmark suite.

| Benchmark | DSR | PQT | VPG | GP | Eureqa | Wolfram |
|---|---|---|---|---|---|---|
| Nguyen-1 | $0.000 \pm 0.000$ | $0.000 \pm 0.000$ | $0.001 \pm 0.004$ | $0.000 \pm 0.000$ | $0.000 \pm 0.000$ | $0.000 \pm 0.000$ |
| Nguyen-2 | $0.000 \pm 0.000$ | $0.001 \pm 0.005$ | $0.021 \pm 0.051$ | $0.001 \pm 0.003$ | $0.000 \pm 0.000$ | $0.000 \pm 0.000$ |
| Nguyen-3 | $0.000 \pm 0.000$ | $0.004 \pm 0.010$ | $0.051 \pm 0.045$ | $0.000 \pm 0.000$ | $0.001 \pm 0.005$ | $0.000 \pm 0.000$ |
| Nguyen-4 | $0.000 \pm 0.000$ | $0.001 \pm 0.004$ | $0.041 \pm 0.020$ | $0.000 \pm 0.000$ | $0.003 \pm 0.006$ | $0.000 \pm 0.000$ |
| Nguyen-5 | $0.015 \pm 0.026$ | $0.007 \pm 0.014$ | $0.062 \pm 0.024$ | $0.005 \pm 0.011$ | $0.001 \pm 0.003$ | $0.802 \pm 0.398$ |
| Nguyen-6 | $0.000 \pm 0.000$ | $0.000 \pm 0.003$ | $0.000 \pm 0.000$ | $0.001 \pm 0.002$ | $0.000 \pm 0.000$ | $0.609 \pm 0.059$ |
| Nguyen-7 | $0.005 \pm 0.006$ | $0.006 \pm 0.007$ | $0.021 \pm 0.010$ | $0.003 \pm 0.001$ | $0.000 \pm 0.002$ | $0.054 \pm 0.000$ |
| Nguyen-8 | $0.003 \pm 0.014$ | $0.048 \pm 0.037$ | $0.125 \pm 0.113$ | $0.072 \pm 0.081$ | $0.075 \pm 0.115$ | $0.006 \pm 0.034$ |
| Nguyen-9 | $0.000 \pm 0.000$ | $0.000 \pm 0.000$ | $0.000 \pm 0.000$ | $0.000 \pm 0.000$ | $0.000 \pm 0.000$ | – |
| Nguyen-10 | $0.000 \pm 0.000$ | $0.004 \pm 0.013$ | $0.001 \pm 0.014$ | $0.006 \pm 0.018$ | $0.006 \pm 0.012$ | – |
| Nguyen-11 | $0.000 \pm 0.000$ | $0.000 \pm 0.000$ | $0.000 \pm 0.000$ | $0.081 \pm 0.135$ | $0.000 \pm 0.000$ | – |
| Nguyen-12 | $0.202 \pm 0.060$ | $0.212 \pm 0.061$ | $0.249 \pm 0.058$ | $0.139 \pm 0.041$ | $0.068 \pm 0.062$ | – |
| Average | $0.019 \pm 0.059$ | $0.023 \pm 0.062$ | $0.048 \pm 0.082$ | $0.026 \pm 0.065$ | $0.013 \pm 0.046$ | – |
| | | | | | | |
| Nguyen-2$'$ | $0.001 \pm 0.006$ | $0.003 \pm 0.010$ | $0.190 \pm 0.071$ | $0.023 \pm 0.101$ | $0.018 \pm 0.020$ | $0.000 \pm 0.000$ |
| Nguyen-5$'$ | $0.002 \pm 0.004$ | $0.022 \pm 0.033$ | $0.221 \pm 0.160$ | $0.022 \pm 0.017$ | $0.011 \pm 0.012$ | $0.901 \pm 0.297$ |
| Nguyen-8$'$ | $0.046 \pm 0.121$ | $0.083 \pm 0.186$ | $0.322 \pm 0.269$ | $0.173 \pm 0.230$ | $0.115 \pm 0.123$ | $0.218 \pm 0.406$ |
| Nguyen-8$''$ | $0.054 \pm 0.028$ | $0.041 \pm 0.027$ | $0.089 \pm 0.023$ | $0.047 \pm 0.068$ | $0.029 \pm 0.038$ | $0.041 \pm 0.089$ |
| Average | $0.025 \pm 0.066$ | $0.036 \pm 0.097$ | $0.204 \pm 0.179$ | $0.066 \pm 0.144$ | $0.042 \pm 0.077$ | $0.290 \pm 0.443$ |
| | | | | | | |
| Nguyen-1$^c$ | $0.000 \pm 0.000$ | $0.000 \pm 0.000$ | $0.000 \pm 0.000$ | $0.000 \pm 0.000$ | $0.000 \pm 0.000$ | $0.000 \pm 0.000$ |
| Nguyen-5$^c$ | $0.000 \pm 0.000$ | $0.000 \pm 0.000$ | $0.000 \pm 0.000$ | $0.000 \pm 0.000$ | $0.000 \pm 0.000$ | $0.714 \pm 0.449$ |
| Nguyen-7$^c$ | $0.000 \pm 0.000$ | $0.001 \pm 0.002$ | $0.000 \pm 0.000$ | $0.011 \pm 0.031$ | $0.004 \pm 0.016$ | $0.043 \pm 0.000$ |
| Nguyen-8$^c$ | $0.000 \pm 0.000$ | $0.000 \pm 0.000$ | $0.000 \pm 0.000$ | $0.005 \pm 0.004$ | $0.002 \pm 0.003$ | $0.045 \pm 0.073$ |
| Nguyen-10$^c$ | $0.000 \pm 0.000$ | $0.000 \pm 0.000$ | $0.004 \pm 0.006$ | $0.002 \pm 0.001$ | $0.000 \pm 0.000$ | – |
| Average | $0.000 \pm 0.000$ | $0.000 \pm 0.001$ | $0.001 \pm 0.003$ | $0.004 \pm 0.014$ | $0.001 \pm 0.007$ | – |

and Wolfram) only provide the final solution, so training curves are not possible. Note that the recovery rate values in Table 1 correspond to the final point on each curve in Figure 7.

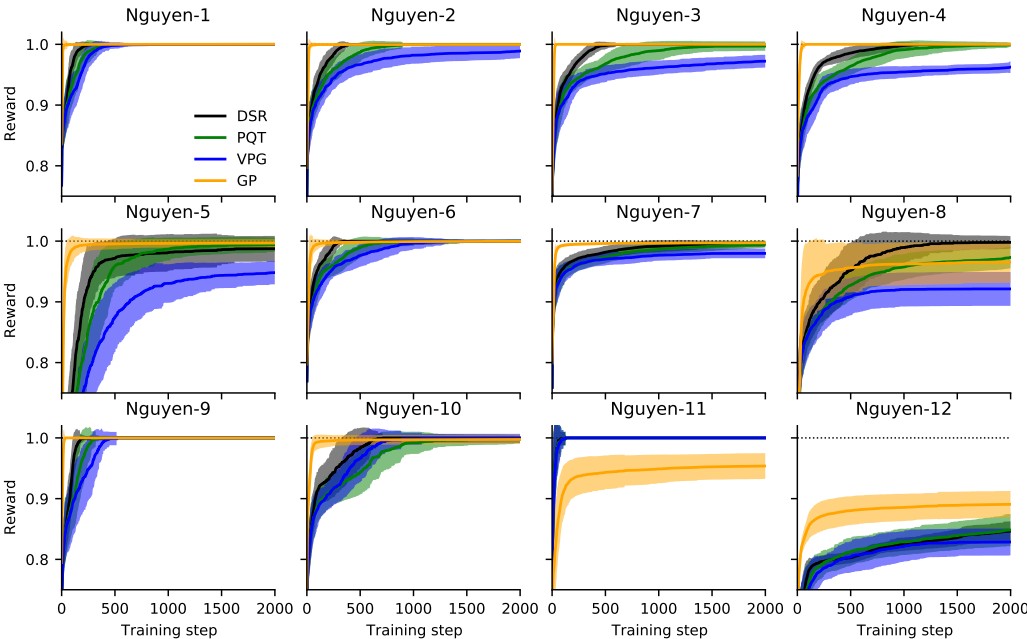

Figure 6: Reward training curves for DSR, PQT, VPG, and GP for the Nguyen benchmarks. Each curve shows the best reward $(1/(1 + \text{NRMSE}))$ found so far as a function of expressions evaluated, averaged across all training runs. A value of 1.0 denotes that all training runs recovered the correct expression. Error bands represent standard deviation.

**Distributions of rewards during training.** Figure 2 compares the performance of the risk-seeking policy gradient and standard policy gradient for Nguyen-8 by showing the distributions of rewards during training. Analogous plots for all 12 Nguyen benchmarks are shown in Figures 8 and 9.

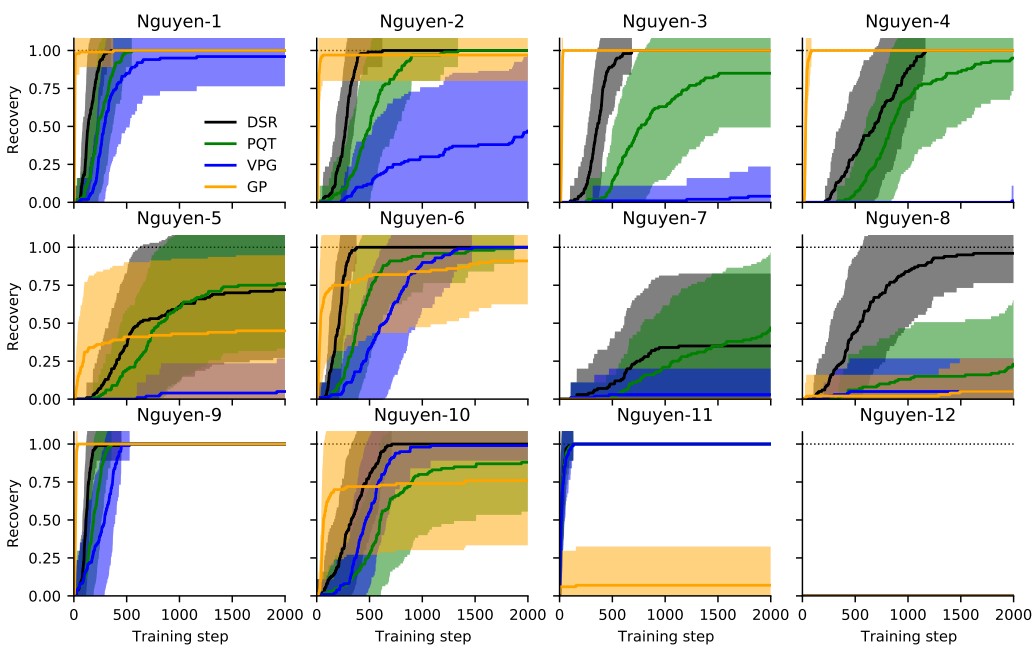

Figure 7: Recovery rate training curves for DSR, PQT, VPG, and GP for the Nguyen benchmarks. Each curve shows the fraction of independent training runs that correctly recovered the benchmark expression as a function of expressions evaluated. A value of 100% denotes that all training runs recovered the correct expression.

## APPENDIX G    RUNTIMES OF NON-COMMERCIAL ALGORITHMS

All experiments used an identical number of total evaluation expressions for each method. Another consideration is the compute time of different methods. In Table 11, we report the average runtimes (on a single core) for non-commercial algorithms across the Nguyen benchmark set. We report two runtime metrics: 1) runtime to process all 2M expressions, without early stopping if the benchmark expression is recovered; 2) runtime until recovering the benchmark expression. Without early stopping, GP is unsurprisingly the fastest method, as it does not require neural network training. However, when considering early stopping, DSR is the fastest, largely because it has the highest recovery rate, thus triggering early stopping more often.

Table 11: Comparison of runtimes across the Nguyen benchmark set.

| Method | Early stopping | |
| --- | --- | --- |
| | No | Yes |
| DSR | $1920.7 \pm 342.9$ s | $483.0 \pm 641.9$ s |
| PQT | $2320.3 \pm 237.8$ s | $959.6 \pm 925.9$ s |
| VPG | $2452.3 \pm 107.4$ s | $1477.1 \pm 1058.7$ s |
| GP | $1375.4 \pm 334.0$ s | $915.5 \pm 433.0$ s |

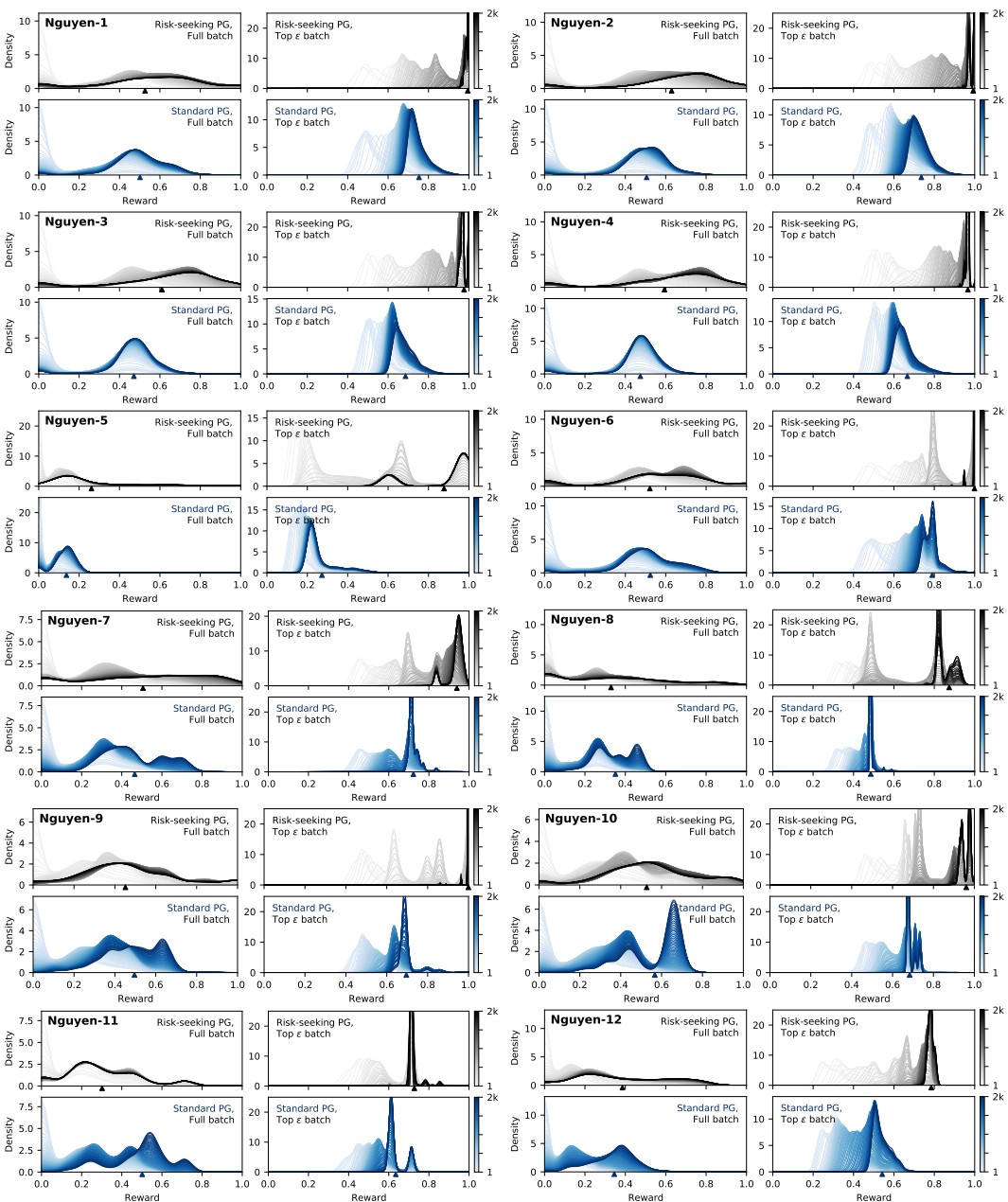

Figure 8: Empirical reward distributions for Nguyen benchmarks, with and without risk-seeking policy gradients. Each group of four plots corresponds to a particular benchmark expression. Each curve is a Gaussian kernel density estimate (bandwidth 0.25) of the rewards for a particular training iteration, using either the full batch of expressions (plots labeled "Full batch") or the top $\varepsilon$ fraction of the batch (plots labeled "Top $\varepsilon$ batch"), averaged over all training runs. Black plots were trained using the risk-seeking policy gradient objective. Blue plots were trained using the standard policy gradient objective. Colorbars indicate training step. Triangle markings denote the empirical mean of the distribution at the final training step.

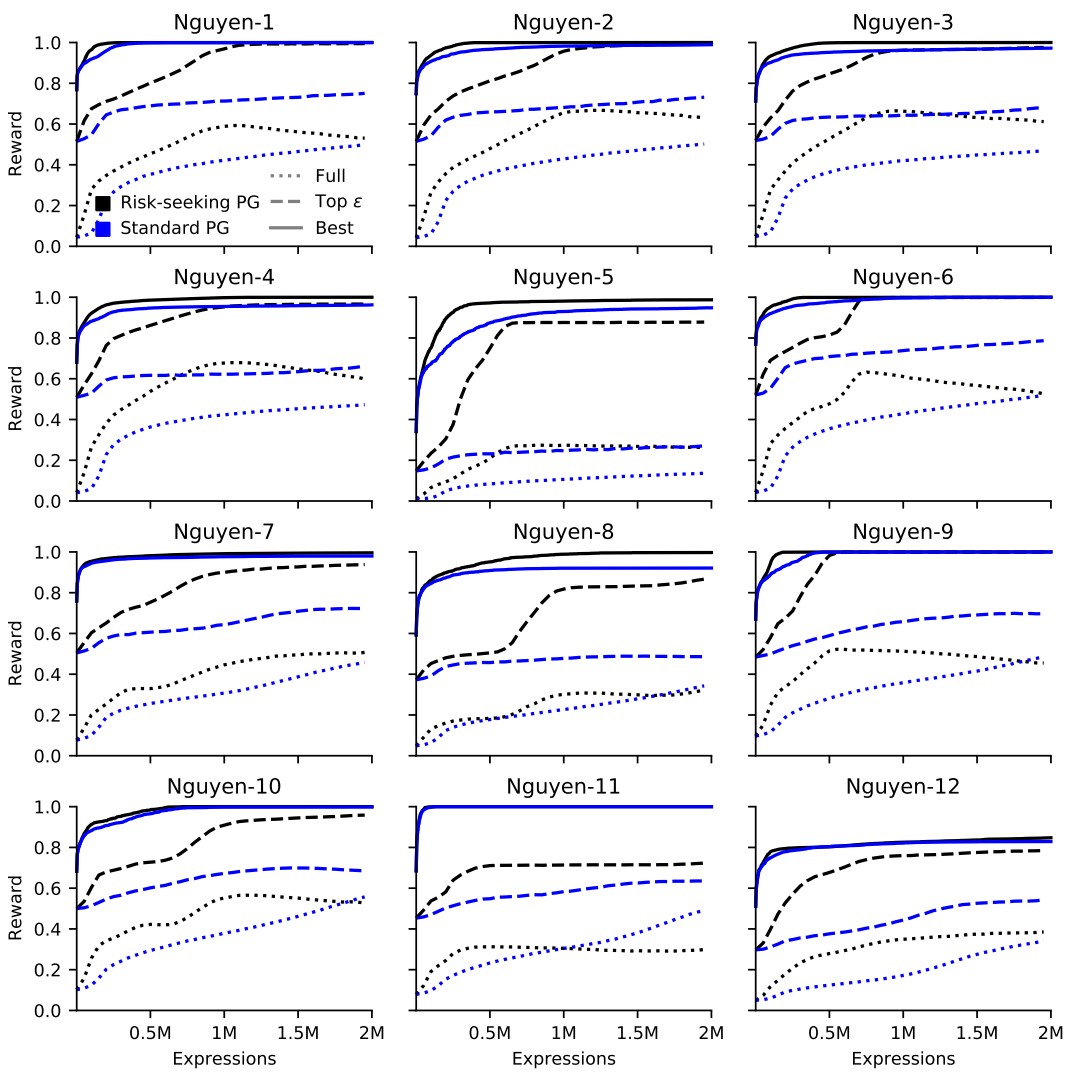

Figure 9: Training curves for all Nguyen benchmarks, with and without risk-seeking policy gradients. Dotted curves denote mean reward of the full batch. Dashed curves denote mean reward of the top $\varepsilon$ fraction of the batch. Solid curves denote the best expression found so far. Each curve is averaged over all training runs.

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
