# OpenReview forum: "Deep symbolic regression: Recovering mathematical expressions from data via risk-seeking policy gradients"
_ICLR.cc/2021/Conference — ICLR 2021 Oral_

### Official Review · AnonReviewer3 · 2020-10-14
**Well-written, well-executed, well-suited, well done**

**Rating:** 9
**Confidence:** 3

**Review:**

The authors introduce a novel method for inferring a simple algebraic expression for an output in terms of an input. The method outputs a sequence tokens of the algebraic expression as represented by the pre-order traversal of an expression tree. For each token of the sequence, a recurrent neural network outputs a probability distribution on possible tokens, from which the token is sampled. Some notable contributions:

1. The authors introduce a novel reinforcement learning objective that optimizes the quality of the best examples instead of optimizing the average quality.

2. At each point in the sequence, their recurrent neural network takes as input the sibling and parent nodes of the expression tree, instead of the previous token in the sequence.

3. Simple constraints can be introduced in their approach by zeroing out the sampling probability for tokens that don't meet the constraints. (For example, they disallow redundant expressions like log(exp(x))).

The authors do a comprehensive comparison with other methods (in which their work performs very favorably) as well as an ablation analysis to confirm that each of their innovations is helpful (which I was very pleased to see).

Well done.

---

> ### Author Response · Authors · 2020-11-18
> **Thank you for your review**
>
> Thank you for the kind words and recognizing our various contributions. The points you highlight concisely summarize the fundamental contributions of our paper.

---

### Official Review · AnonReviewer2 · 2020-10-26

**Rating:** 8
**Confidence:** 4

**Review:**

Summary

The paper uses RNN trained by risk seeking RL objective to predict mathematical expression that generated the target dataset. In the second step placeholders for constants in sampled expression are optimized again by gradient optimizer to maximize the reward function. The decoder RNN uses heuristics that make it easier to generate valid math expressions.


Strong points

+ The paper is clearly written and potentially relevant to larger audience.
+ The proposed method is relatively easy to implement and it leads to good empirical results.


Recommendation

I recommend acceptance of this paper since the proposed technique is highly competitive, easy to implement and well presented.


Questions

* How does compute used by GP to get results in Tab 1 compare to policy gradient based methods? My perspective is that all the techniques should find the correct solution in the limit with enough computational resources (under assumption that they can escape local optima). Therefore making sure that all the methods had roughly similar budgets is important.

* In figure 2E mean reward for standard PG is going up even after 2M training steps, would it be possible to run the experiment longer to see where the max would be once mean reward converges? (Similarly in figure 8 in tasks Nguyen 3,5 and 8 mean of standard GP is still going up.) It is great to see that risk seeking PG converges much faster on these tasks, however knowing what is the limit for standard PG would be also interesting.

* In section 3.1 you say that generated expressions had to be between 4 and 30 symbols long, however tasks 8 and 11 (sqrt(x) and x^y) have shorter descriptions. What am I missing?


Possible improvements

* Using the set of problems from AI Feynman would make empirical evaluation stronger. However the same can be said for AI Feynman system and Nguyen dataset.

* Beyond scope of this paper: At the moment a new sequential model is learned for each task and it learns about that task only through the reward function. What if the model can "see" the dataset first by reading it through n-dimensional CNN (or RNN) that would produce "dataset embedding" that will be later used to condition the sampling RNN. In the same way that image captioning model is conditioned on image embedding. Generating infinite training dataset (with random expressions) should be trivial, that is one advantage over image captioning with limited data.

Typos

standard standard -> standard

---

> ### Author Response · Authors · 2020-11-13
> **Response to Reviewer 2 (1 of 2)**
>
> Thank you for your insightful review, feedback, and ideas for future work!
> _____
>
> **[Compute time of GP vs DSR]** We provide runtimes for GP and the three policy gradient-based methods, which we add in the new Table 11. We present two methods to compare runtimes: Method 1 reports the time to run each algorithm for the maximum of 2M total expressions (i.e. no early stopping if the solution is recovered); thus, it measures training speed without taking into account recovery performance. Method 2 reports time to recover the true expression (up to the 2M maximum). The average and standard deviation for run times on the Nguyen benchmarks using these two methods are:
>
> | Algorithm | $\hphantom{gap}$Runtime (Method 1) | $\hphantom{gap}$Runtime (Method 2)
> | :---: | :---: | :---: |
> | DSR | $1920.7 \pm 342.9$ s | $483.0 \pm 641.9$ s |
> | PQT | $2320.3 \pm  237.8$ s | $959.6 \pm 925.9$ s |
> | VPG | $2452.3 \pm 107.4$ s | $1477.1 \pm 1058.7$ s |
> | GP | $1375.4 \pm 334.0$ s | $915.5 \pm 433.0$ s |
>
> In general, the runtimes for policy gradient-based methods using Method 1 are all similar because only the objective function differs, otherwise the entire framework is shared. GP is slightly faster than these because there is no neural network overhead. However, reward computation is a computational bottleneck, hence the small overall differences across algorithms. Using Method 2, DSR outperforms GP, largely due to finding the solution in fewer expression evaluations.
> _____
>
> **[Similar budgets across algorithms]** You are correct that providing similar budgets across all baselines is important for fair comparison. Two reasonable metrics to assess budgets are 1) total number of expression evaluations (i.e. sample efficiency) and 2) wall time (i.e. computational efficiency). We elected to compare sample efficiency by limiting all algorithms to 2M expression evaluations (except the two commercial software methods, Eureqa and Wolfram, which are not transparent about number of expression evaluations and thus we instead run to completion.) The primary reason we chose to fix number of samples (as opposed to wall time) was precisely because all algorithms converge to local optima, typically by 2M expressions (except for a few cases, as you point out, which we address below).
>
> To expand on this point, perhaps surprisingly, it is actually *not* the case that these methods (whether GP-based or neural network-based) will identify the correct solution given enough computational budget. That is, they *do* get stuck in local optima as you suggested. GP approaches have a difficult time escaping local optima, especially when the global optimum looks quite symbolically different, e.g. converging on a Taylor expansion of an expression when the true expression uses lots of trigonometric operators (Turner and Miller, 2015). Policy gradient algorithms are also known to converge to local optima $-$ often quite prematurely $-$ (Wang et al., 2019). Indeed, during the course of this study we often observed policy gradient algorithms converging on a small number of samples (i.e. the RNN samples the same few expressions over and over).
> _____
>
> **[Convergence of VPG on Nguyen-3, 5, and 8]** As requested, we performed experiments for 5x longer using standard policy gradient (VPG) with 10,000 training steps (10M sampled expressions) on benchmarks Nguyen-3, 5, and 8, which had not quite converged in the main experiments. Please see the added plots in the Supplementary file "VPG-convergence.pdf." For Nguyen-5 and Nguyen-8, the expected reward by the end of training converges to sub-optimal values (0.6 and 0.48, respectively) and the exact expressions (global optima) are not recovered (i.e. the best reward $<1$). For Nguyen-3, the expected reward of the policy eventually converges to the optimal value and the exact expression (global optima) gets sampled (best reward $=1$). This diverse behaviour highly depends on the characteristics of the problem at hand, e.g., it is known to depend on the locality of the representation in the discrete space (Galv\'an-L\'opez et al., 2010), and is a reason why symbolic regression is such a difficult problem.
>
> Additionally, as a diagnostic metric we measure the number of new expressions found each training step (i.e., expressions that have never been seen during a particular training run). In the above experiments, VPG produced no new expressions after 9M samples, confirming that the search converged.

---

> > ### Author Response · Authors · 2020-11-13
> > **Response to Reviewer 2 (2 of 2)**
> >
> > (continued)
> >
> > **[Token sequences for $\sqrt{x}$ and $x^y$]** This is an excellent question and is clarified in the original submission, although it was somewhat buried (and thus easy to miss) in Appendix D and so we expand upon it here. The square root and power operators are not part of the function set as specified by the Nguyen benchmarks; this is precisely what makes these benchmark tasks challenging. However, these benchmarks can still be recovered with the given function set via alternate algebraic manipulation. In particular, Nguyen-8 can still be exactly recovered, e.g. via the sequence $[\exp, \times, \div, x, +, x, x, \log, x]$, which translates to $\exp(\frac{x}{x+x}\log(x))$ and simplifies to $\sqrt{x}$ (noting $x>0$). Similarly, Nguyen-11 can still be recovered via the sequence $[\exp, \times, y, \log, x]$, which translates to $\exp(y\log(x))$ and simplifies to $x^y$.
> > _____
> >
> > **[AI Feynman benchmarks]** Designing benchmarks for symbolic regression is a challenging activity and known struggle within the field (McDermott et al., 2012). In particular, it is often quite unintuitive what makes a particular expression a challenging (or not) symbolic regression problem. Thus, we chose to stick with well-established, community-vetted benchmarks like the Nguyen benchmark set that have *many* existing works to establish a baseline for difficulty.
> >
> > While we do appreciate that one contribution of AI Feynman is introducing a new symbolic regression benchmark suite, we are also wary of adopting it due to some peculiarities. For example, the datasets treat physical constants (like the universal gravitational constant $G$) as variables with arbitrary domains. Also, many of the problems can be solved with dimensional analysis alone; in practice, a real-world practitioner of symbolic regression would apply dimensional analysis as a pre-processing step to simplify the problem before employing a symbolic regression algorithm.
> > _____
> >
> > **[Suggestion for future work]** Thank you for the valuable suggestion! This would be very reminiscent of image-to-caption systems (Vinyals et al., 2015), but instead mapping datasets to expression traversals. A similar research direction we are pursuing is using supervised learning on input datasets (and their corresponding expression traversals) to learn a *prior* over which tokens may exist in that dataset. Even just learning a prior over the first token (i.e. outermost operator) of a dataset's expression could be a powerful, data-driven way to reduce the search space for symbolic regression. Such priors can be incorporated into our framework by adding logits directly to the logits emitted by the RNN (just like how we impose in situ constraints by adding $-\infty$ to the logits of tokens that would violate a constraint).

---

> > > ### Comment · AnonReviewer2 · 2020-11-23
> > > **Insightful responses**
> > >
> > > Thank you for your answers. It is nice to see Table 11 with compute times. I would add results from VPG-convergence.pdf to the appendix as well.

---

### Official Review · AnonReviewer4 · 2020-10-26
**Accept**

**Rating:** 7
**Confidence:** 4

**Review:**

#### Summary

This paper presents a symbolic regression algorithm which uses policy gradient to learn a distribution over the space of mathematical expression structures. The distribution is represented by an RNN, and the on-policy sampling is also realised by this RNN. To prune the massive sample space, the authors include several constraints according to some prior domain knowledge. Furthermore, instead of optimising the RNN directly with policy gradient, the proposed algorithm also makes use of the risk-constrained method that emphasises the risks above a given percentile criteria. The proposed approach has achieved the best performance on several benchmark symbolic regression datasets.

#### Pros

+ This paper is motivated by an interesting problem. I like the idea of using domain knowledge to introduce hierarchical constraints, which seems to be effective in this task.
+ The authors have done extensive experiments, the presented algorithm performs well on the benchmark dataset.
+ This paper has covered a wide range of related work.

#### Cons

- The main contribution of this paper is introducing the heuristic constraints to prune the expression search space and applying existing reinforcement learning techniques (e.g., using RNN to represent the policy of sampling a grammar; the risk-constrained reinforcement learning) to the symbolic regression task. It looks more incremental rather than a novel contribution.
- The number of experiments in the main article is small, and many experimental details are hidden in the supplementary, I think the authors should re-organise the paper and prompt some of the results in the appendix to the main article.
- One of the biggest problem for the on-policy reinforcement learning is its time complexity since it requires a large sample for estimating the expectation of risk. Considering that symbolic regression is a hard problem with highly complicated underlying distribution over the expression structures, I wonder how efficient is the proposed approach comparing to the other baseline methods. I think the authors should include the results of running time of all the compared approaches.
- The experimental setting is questionable for some compared methods. In Appendix E, the authors report the results of several SOTA symbolic regression approaches. However, as the authors stated, the experimental results are different because the choice of datasets and primitive functions are different. I have noticed that this is a re-submission from ICLR 2020, and reviewers were pointing out the problem of lacking comparison one year ago. I think one year should be enough for carrying out experiments of all these SOTA approaches under the same experimental setting. Still, there is no such result in the current submission.

#### Recommendation

I think this paper proposes a nice approach to tackle the symbolic regression problem. Although the presented algorithm seems to be a combination of existing approaches, its performance on benchmark data sets looks quite good. However, the experiments are still weak because this paper lacks a fair comparison (e.g., using same library and the same set of problems), and my biggest concern of the proposed approach is its time complexity. I think the authors should address these problems to improve this work.

--------------------------------------------------------------------------
#### Update after rebuttal
The authors have explained the reasons for the comparing experiments in Appendix E and updated the result of time complexity, which I think worth including in this paper.

---

> ### Author Response · Authors · 2020-11-12
> **Contributions, Time complexity, and Number of experiments**
>
> Thank you for your detailed review of our work! Please see our responses below.
>
> [Contributions] We appreciate recognizing the ability to flexibly impose in situ constraints as a valuable contribution. We maintain that another important contribution is developing the risk-seeking policy gradient. While risk-constrained reinforcement learning has many important works (e.g. Rajeswaran et al. 2016, Tamar et al. 2014, Chow et al. 2017), these all discuss *risk-averse* strategies. Risk-averse strategies are commonly used in engineering and finance to avoid policies that can lead to catastrophic trajectories with prohibitively large costs, e.g. avoiding a policy that can result in a robot smashing against an obstacle $-$ even if the policy has a large expected reward. Here, we propose something different: a strategy to promote policies that can produce outstanding trajectories via a "high risk, high reward" objective, even if the policy has lower expected reward. To our knowledge, ours is the first proposed risk-seeking policy gradient. While the theoretical formulations are similar (as we demonstrate), risk-seeking in discrete optimization is quite a different problem setting because the ultimate goal is to identify the *single best* discrete object. Essentially, we use it as a proxy for the "true" objective to find $\tau^\star = \arg\max_{\tau}R(\tau)$, which is clearly intractable without enumerating over the entire search space.
>
> Priority queue training (PQT; Abolafia et al. 2018), which is one of our most important baselines, also addresses the challenge of finding the *best* discrete object; however, it is limited to RL environments with deterministic transition dynamics and (as demonstrated in Figure 5) can catastrophically fail when the reward function is stochastic. Our proposed risk-seeking policy gradient is more general, and can be applied to any RL environment to maximize best-case rewards (e.g. to achieve a new "high score" in Atari).
>
> [Time complexity] As requested, we provide runtimes for DSR and baselines, which we add in the new Table 11. We identify two methods to compare runtimes: Method 1 reports the total time to run each algorithm for the maximum of 2M total expressions (i.e. no early stopping if the solution is recovered); thus, it measures training speed without taking into account recovery performance. Method 2 reports total runtime before recovering the true expression (up to the 2M maximum).
>
> The average and standard deviation for runtimes on the Nguyen benchmarks using these two methods are:
>
> | Algorithm | $\hphantom{gap}$Runtime (Method 1) | $\hphantom{gap}$Runtime (Method 2)
> | :---: | :---: | :---: |
> | DSR | $1920.7 \pm 342.9$ s | $483.0 \pm 641.9$ s |
> | PQT | $2320.3 \pm  237.8$ s | $959.6 \pm 925.9$ s |
> | VPG | $2452.3 \pm 107.4$ s | $1477.1 \pm 1058.7$ s |
> | GP | $1375.4 \pm 334.0$ s | $915.5 \pm 433.0$ s |
>
> In general, the runtimes for DSR, VPG, and PQT using Method 1 are all similar because only the objective function differs, otherwise the entire framework is shared. GP is slightly faster than these because there is no neural network overhead. However, reward computation is a computational bottleneck, hence the small overall differences across algorithms.
>
> Using Method 2, DSR outperforms even GP, largely due to finding the solution in fewer expression evaluations.
>
> [Number of experiments] We note that the number of experiments we performed is extremely high compared to any other baseline we identified. Namely, we performed $n=100$ independent runs of our experiments, which we believed important to demonstrate statistical significance, especially for difficult benchmarks with low recovery rate. Thus, Table 1 alone consists of 6800 independent experiments, 4800 of which require training a neural network. In total, 38920 experiments were performed to produce the results in the paper. Most other (stochastic) symbolic regression methods report far fewer replicates. Thus, within our overall computational resource budget across all experiments, we prioritized statistical power over a larger set of benchmarks.

---

> ### Author Response · Authors · 2020-11-12
> **Experimental setting of Appendix E results**
>
> [Experimental setting in Appendix E] We first clarify two points regarding the additional baseline experiments in Appendix E, then discuss why we directly compared to their literature-reported results. First, the baselines in Appendix E are not SOTA methods in symbolic regression (nor do they claim to be). We point this out because we believe that the empirical evaluation of DSR would be adequate without them, i.e. with only the baselines included in Table 1. The intention of these additional experiments was partially to proactively allay potential concerns like "What about comparing to method $\rule{1cm}{0.15mm}$?" We preemptively addressed this possible concern by comparing to an additional smattering of interesting but non-SOTA symbolic regression baselines, i.e. those in Appendix E. In contrast, we reserved the most important baselines for Table 1 experiments: those that are SOTA in symbolic regression (Eureqa), SOTA in neural-guided discrete optimization (PQT), or other widely adopted generic baselines (VPG, GP).
>
> Second, and more importantly, we stress that for each of these additional baselines in Appendix E, we *exactly reproduced* their experimental setup, i.e. by using identical total number of expressions, number of independent trials, train/test datasets, primitive sets, measures of performance, etc. We carefully recapitulated each baseline's experimental design decisions, to ensure a fair comparison. Thus, we confidently reassure all readers that all experiments (both in Table 1 and Appendix E) are fair comparisons to their respective baselines.
>
> For Appendix E baselines, the primary reason we directly compare to literature-reported results (rather than use these baselines to mimic our Table 1 experimental setup) is to avoid several sources of human bias. One source of bias can come from implementation: it is easy to inadvertently introduce errors via software bugs and/or misinterpreting aspects of the method. Further, not all references provide sufficient information to reproduce the method. Even when a baseline's software is open-sourced, it can be rigid in its ability to adapt to a different experimental setting, requiring modifications that introduce opportunities for bias. Another source of bias is in selecting experimental setup design choices, e.g. making design decisions that are beneficial to one method more than another.
>
> Instead, by recapitulating each baseline's experimental setup, and directly comparing to reported results at face value, we demonstrate that we can "beat them at their own game," so to speak. In other words, we give each baseline the benefit of the doubt that details of their experimental setup, e.g. total number of expressions to evaluate, are appropriate for (or even overfit to) their particular method. By mimicking their setup, we eliminate the possibility of an unfair advantage. Thus, we believe these Appendix E results, in conjunction with the widely used Nguyen benchmark set and SOTA baselines in Table 1, provide strong empirical justification that DSR is SOTA in recovering benchmark expressions.
>
> Please let us know if there is anything else we can clarify to assure readers that comparisons are fair across all experiments.

---

> ### Comment · Area_Chair1 · 2020-11-23
> **How does the responses from authors change or not change your evaluation?**
>
> Dear AnonReviewer4,
>
> Thank you very much for your review.  You have raised several concerns about this paper.  Your criticisms seem quite valid, but the authors have provided detailed responses to them.  How does your evaluation change or not change after reading those responses?

---

> > ### Comment · AnonReviewer4 · 2020-11-23
> > **Update**
> >
> > Dear AC,
> >
> > The authors have answered my questions and reported the result on time complexity, which looks good. I'll update my review and rating.

---

### Official Review · AnonReviewer1 · 2020-10-28
**Recurrent neural networks trained in a reinforcement learning framework for symbolic regression**

**Rating:** 8
**Confidence:** 3

**Review:**

This paper presents a novel approach to the problem of symbolic regression, where the goal is to learn relationships between variables in the form of mathematical expressions. This is clearly a very relevant task towards constructing explainable AI systems.

The proposed approach in this paper is based on the generation of mathematical expression with recurrent neural networks, by exploiting background knowledge about the form of the expressions to impose constraints on the generated examples.

The RNN is trained by maximizing a risk-seeking policy gradient that aims to increase best-case performance. The key idea here is to increase the reward of the top-epsilon fraction of samples from the distribution, without taking into account the samples that fall below such threshold.

The technique is sound and novel, and it provides a significant contribution to this research area. The positioning of the paper with respect to the state-of-the-art in the field is highly accurate.

A very solid experimental evaluation is carried out on several benchmarks, comparing the proposed approach with state-of-the-art systems for the same task, including commercial software such as Eureqa and Wolfram. The analysis includes an ablation study and experiments conducted with different amounts of training data. The proposed methodology is shown to perform better than all the competitors.

Overall, I consider the paper to be a strong contribution for ICLR.

* In the experiments with different levels of noise in data, why was Gaussian noise added to the dependent variable only, and not also to the independent variables?

- Pag. 4, "is not allowed.While" -> "is not allowed. While"
- Pag. 5, "but in practice has high variance" -> "but in practice it has high variance"

---

> ### Author Response · Authors · 2020-11-17
> **Response to Reviewer 1**
>
> Thank you for your time and interest in reviewing our work.
> _____
>
> **[Adding noise to independent vs dependent variable.]** We added Gaussian noise to the dependent/response variable because it is more relevant to real-world, experimental settings. First, practitioners often have high precision controlling input variables but experience measurement error when measuring the response. For example, in materials science, temperature and applied strain may be input variables, which can be controlled very accurately. However, measurement of output variables, e.g. stress, is limited by sensor technology. Second, many domains (e.g. biology) exhibit stochasticity in the underlying system's response to inputs. For example, the amount of product formed in a biochemical reaction given fixed input concentrations is an inherently stochastic variable. This too would be manifested in uncertainty in the output variable.
> _____
>
> **[Typos.]** Thank you for pointing out the typos; we have fixed them in our revised submission.

---

> > ### Comment · AnonReviewer1 · 2020-11-23
> > **Thanks**
> >
> > Thank you very much for the clarification.

---

### Decision · Program_Chairs · 2021-01-07
**Final Decision**

**Decision:**

Accept (Oral)

**Comment:**

This paper proposes an approach of generating mathematical expressions with a recurrent neural network, which is trained with risk-seeking policy gradient to maximize the quality of best examples rather than average examples.  The proposed approach also enables easily incorporating domain knowledge or constraints to avoid illegal or redundant expressions.  In extensive experiments, the proposed method is shown to significantly outperform strong baselines, including commercial software.  All of the reviewers find the work interesting and relevant, and there are no major concerns or issues after discussion.  The topic is also of interest to a wide range of audience in the ICLR community.